# Integrative multi-omics data provide insights into the biosynthesis of furanocoumarins and mechanisms regulating their accumulation in *Angelica dahurica*

Jiaojiao Ji[1,6], Xiaoxu Han[1,2,6], Lanlan Zang[1], Yushan Li[3], Liqun Lin[1], Donghua Hu[1,4], Shichao Sun[1], Yonglin Ren[2], Garth Maker [2], Zefu Lu [3] ✉ & Li Wang [1,5] ✉

Furocoumarins (FCs), important natural compounds with biodefense roles and pharmacological activities, are notably abundant in medicinal plant *Angelica dahurica*. However, its accumulation patterns over development stages in FC-enriched tissue, biosynthetic pathways, and regulatory mechanisms in *A. dahurica* remain elusive. Here, we quantified the concentration dynamics of 17 coumarins across six developmental stages of root and found a gradual decrease in FC concentration as the roots develop. Using a de-novo assembled chromosome-level genome for *A. dahurica*, we conducted integrative multi-omics analyses to screen out candidate genes to fill in the sole missing step in the biosynthesis of imperatorin and isoimperatorin. This revealed that CYP71AZ18 catalyzes hydroxylation at the *C*-5 position of psoralen to generate bergaptol, while CYP71AZ19 and CYP83F95 catalyze hydroxylation at the *C*-8 position to produce xanthotoxol, notably indicating that a single step is catalyzed by two genes from distinct CYP450 subfamilies in this species. CYP71AZ19 originated from a proximal duplication event of CYP71AZ18, specific to *A. dahurica*, and subsequently underwent neofunctionalization. Accessible chromatin regions (ACRs), especially proximal ACRs, correlated with high gene expression levels, and the three validated genes exhibited strong signals of ACRs, showing the importance of chromosomal accessibility in regulating metabolite biosynthesis.

Plants have evolved the ability to produce specialized metabolites as an adaptive response to environment. Among these metabolites, furocoumarins (FCs) are defense chemicals against various bioaggressors. FCs play a crucial ecological role by defending against pathogens, inhibiting germination of competing plants, and deterring herbivores[1]. Additionally, FCs exhibit substantial pharmacological efficacy, encompassing anticancer, antimicrobial, and anti-inflammatory effects in human[2,3]. FCs were exclusively identified in a few phylogenetically distant plant families, including the Apiaceae, Rutaceae, Fabaceae and Moraceae. The sporadic FCs distribution across angiosperm phylogeny implies multiple independent origins of FC biosynthesis in the four families[4,5].

Despite decades of study, the biosynthesis pathway of FC remains incomplete. It involves prenyltransferases (PTs), O-methyltransferases (OMTs) and cytochrome P450s (CYP450s)[5–9]. CYP450s, the most diversified enzyme family in plants[10,11], exhibit unique characteristics, including substrate promiscuity (capacity to convert multiple substrates), catalytic promiscuity (capacity to catalyze different reactions or oxidations at different moiety of the same substrate), and multifunctionality (capacity to

[1]Shenzhen Branch, Guangdong Laboratory of Lingnan Modern Agriculture, Key Laboratory of Synthetic Biology, Ministry of Agriculture and Rural Affairs, Agricultural Genomics Institute at Shenzhen, Chinese Academy of Agricultural Sciences, Shenzhen, China. [2]College of Environmental and Life Sciences, Murdoch University, Murdoch, WA, Australia. [3]Institute of Crop Sciences, Chinese Academy of Agricultural Sciences, Beijing, China. [4]College of Plant Science and Technology, Huazhong Agricultural University, Wuhan, China. [5]Kunpeng Institute of Modern Agriculture at Foshan 528000, Foshan, China. [6]These authors contributed equally: Jiaojiao Ji, Xiaoxu Han. ✉e-mail: luzefu@caas.cn; wangli03@caas.cn

perform a cascade of oxidations)[12]. To date, CYP450s involved in FC biosynthesis belong to the CYP71 clan and are responsible for furan-ring formation and hydroxylation, leading to the diversification of FCs (Fig. 1)[1,8,13–20]. A single P450 enzyme can catalyze different steps. For instance, AsOC, the CYP736A subfamily catalyze in the Apiaceae plant *Angelica sinensis*, both the conversion of 7-demethylsubersin to marmesin and the conversion of osthenol to columbianetin[19], suggesting its multifunctionality. Noteworthy, FCs are catalyzed by different families of CYP450 in distant plant lineages. For instance, the conversion of xanthotoxin to 5-hydroxyxanthotoxin is catalyzed by CYP71AZ[14], CYP82D[18] and CYP71B[21] families in Apiaceae, Rutaceae, and Moraceae, respectively. Similarly, the marmesin synthetase identified in Apiaceae and Moraceae were from different CYP450 families: CYP736A[19] and CYP76F[1]. Such pattern was not observed within a single plant family. The above evidence underscored the independent evolution of FC biosynthesis.

*Angelica dahurica* is an essential traditional Chinese medicine (TCM) that belongs to the Apiaceae (Chinese Pharmacopoeia Commission, 2020), renowned for its medicinal and edible qualities[22,23]. At least 153 FCs are reported in *A. dahurica* and FCs are the main bioactive compounds for its therapeutic efficacy, which makes *A. dahurica* an ideal system for investigating biosynthetic pathway and its epigenetic regulation of FCs[24]. Imperatorin and isoimperatorin are the quality control standards of *A. dahurica* in

Chinese Pharmacopoeia[25]. Their biosynthesis precursors, bergaptol and xanthotoxol, are products derived from hydroxylation of psoralen at the *C*-5 (catalyzed by psoralen-5-hydroxylase, P5H) and *C*-8 positions (catalyzed by psoralen-8-hydroxylase, P8H), respectively. To date, only one P8H, CYP71AZ4, has been revealed in *Pastinaca sativa*[14], whereas P5H remains unknown in the plant kingdom. Remarkably, P5H is the sole uncharacterized enzyme within the biosynthetic pathway leading from phenylalanine to imperatorin and isoimperatorin in plants[5,6,26,27]. On the other hand, while *A. dahurica* holds significance in TCM, there exists a paucity of research concerning the fluctuations of FCs across its developmental stages[28,29]. This absence hindered the scientific explanation of the harvesting time for *A. dahurica*.

On the other hand, how FC biosynthesis is regulated by epigenetic modifications is obscure. Chromatin accessibility and modification is a hallmark of regulatory DNA[30]. Accessible chromatin regions (ACRs) reflect the gene regulatory capacity, intimately connected to gene expression patterns and metabolite accumulation[31,32]. The chromatin accessibility landscape exhibits tissue-specific variation and undergoes dynamic changes in response to both external environmental cues and internal developmental signals[33]. The assay for transposase-accessible chromatin with high throughput sequencing (ATAC-seq) has emerged as a powerful technology for profiling open chromatin regions across a wide range of species[34,35].

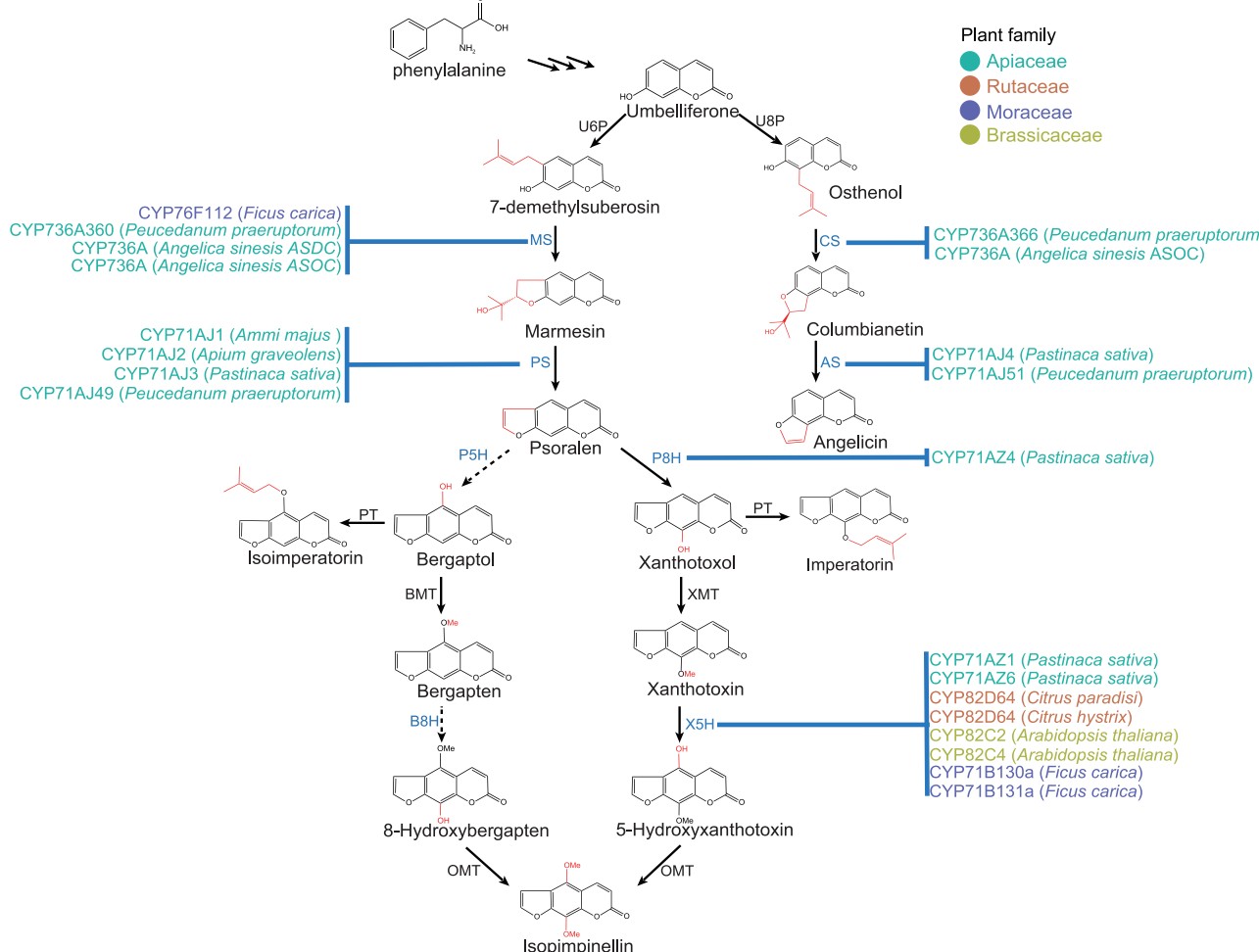

**Fig. 1 | Schematic representation of the furanocoumarin biosynthetic pathway and the validated catalytic enzymes in the Apiaceae, Moraceae, Rutaceae, and Brassicaceae families.** Solid arrows indicate steps for which a biosynthetic gene has been discovered, and dashed arrows indicate proposed step reactions and the catalyzing enzymes are unknown. The red color shows the newly formed structures at each step. U6P umbelliferone-6-prenyltransferase, U8P umbelliferone-8- prenyltransferase, MS marmesin synthetase; CS columbianetin synthetase, PS psoraien synthetase, AS angelicin synthetase, P5H psoraien-5-hydroxylase, P8H psoraien-8-hydroxylase, PT prenyltransferase, BMT bergaptol O-methyltransferase, XMT xanthotoxol O-methyltransferase, B8H bergapten-8-hydroxylase, X5H xanthotoxin-5-hydroxylase, OMT O-methyltransferase.

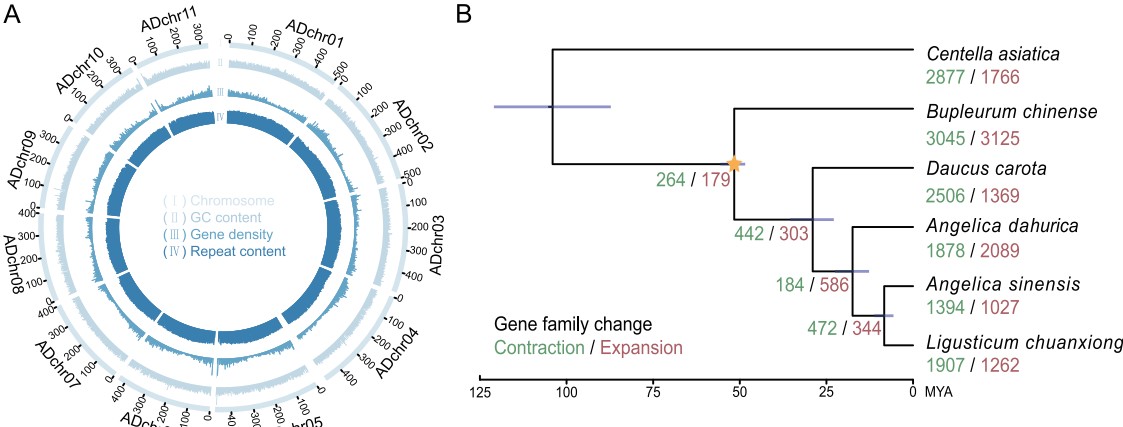

**Fig. 2 | Genomic features and phylogeny of *A. dahurica*. A** Circos plot of genomic features. I pseudochromosomes; II GC content; III gene density; IV repeat content. **B** Inferred phylogenetic tree based on orthologous single-copy genes among six Apiaceae species. Divergence time (million years ago, MYA) was indicated for each node. The constraints for molecular clock calibration are marked with the yellow star. The number of contracted and expanded gene families was marked with green and red digitals, respectively.

Previous studies on chromatin accessibility in plants have primarily focused on few model plants and crops, such as *Arabidopsis thaliana*[36], rice[37], maize[38], and wheat[39]. However, such research has been limited in medicinal plants and, in particular, how ACRs affect the accumulation of secondary metabolites is largely unexplored. Notably, the regulatory function of ACRs in artemisinin biosynthesis has been initially established in *Artemisia annua*[40], a significant discovery that not only unveils the potential role of ACRs in medicinal plants but also underscores the necessity and urgency of delving deeper into their regulatory mechanisms. It is pertinent and imperative to investigate the epigenetic regulatory mechanisms governing secondary metabolite accumulation in medicinal plants through ACRs.

In this work, we present a chromosome-level genome of *A. dahurica* based on PacBio CCS and high-throughput chromosome conformation capture (Hi-C) sequencing data. We generated a comprehensive map delineating the content dynamics of 17 coumarins throughout the root developmental stages of *A. dahurica*. By integrating transcriptomic and metabolomic datasets, our study identified and validated two P8H and one P5H involved in FC biosynthesis. Additionally, we delved into elucidating the regulatory influence of ACRs on gene expression and FCs accumulation. This study completed the biosynthetic pathway from phenylalanine to imperatorin and isoimperatorin, shed light on the evolutionary history of P5H and P8H, and investigated the complex interplay between chromatin accessibility and biochemical traits.

## Results
### *A. dahurica* genome assembly and annotation
We used PacBio sequencing in high fidelity (HiFi) with Hi-C data to assemble the genome (Supplementary Table 1). Assembly of the HiFi reads yielded an initial genome of 4.89 Gb for *A. dahurica*, consistent with flow cytometry and *k*-mer analysis (Supplementary Fig. 1-2). The assembled scaffolds are mapped onto 11 pseudochromosomes (Fig. 2A; Supplementary Table 2-4). The *A. dahurica* genome size was the highest among the published Apiales assemblies[24,41,42]. The assembly achieved 97.20%. Viridiplantae conserved genes for Benchmarking Universal Single-Copy Orthologs (BUSCO) evaluation, and LTR assembly index (LAI) was 20.81, suggesting a gold-level genome (Table. S2). Meanwhile, 81.51–96.15% of RNA-seq data were successfully mapped to the assembled genome (Supplementary Table 3). All the above evidence indicates a high-fidelity genome assembly.

By combining ab initio, homology-based and transcriptome-based approaches, we predicted 61,419 protein-coding genes (Supplementary Table 5). The annotated genes exhibited high completeness with 91.00%

from BUSCO, and 89.04% of them were functionally annotated through at least one of the databases NR, Pfam, Gene Ontology (GO) or Kyoto Encyclopedia of Genes and Genomes (KEGG) (Tables S5), reflecting a high level of annotation completeness. The average sequence lengths for exons and introns are 264.69 and 606.68 bp, respectively (Supplementary Table 5). In total, 4.26 Gb (87.08% of the genome) was identified as repetitive sequences, with long terminal repeat retrotransposons (LTR-RTs) being the predominant category (3.80 Gb, 77.71% of the genome) (Supplementary Table 6).

### Secondary metabolic pathways were enriched in duplicated genes within the expanded gene families of *A. dahurica*
To further investigate the genomic characteristics of *A. dahurica*, we conducted comparative genomic analysis with six other Apiaceae species, including *Centella asiatica*, *Bupleurum chinense*, *Daucus Carota*, *Ligusticum chuanxiong* and *A. sinensis*. A phylogenetic tree of the six species based on orthologous single-copy genes revealed that *A. dahurica* diverged from *D. carota* at approximately 17.45 MYA (Fig. 2B). The phylogenetic topology is mostly consistent with previous reports[41,43].

Gene family contraction and expansion analyses unveiled that 1878 and 2089 gene families contracted and expanded in the *A. dahurica* genome, respectively (Fig. 2B). The number of expanded gene families in *A. dahurica* was almost twice as that in *D. carota*, *A. sinensis* and *L. chuanxiong* (Fig. 2B). Next, genes within the expanded families in *A. dahurica* were classified into five types, whole-genome duplication (WGD), tandem duplication (TD), proximal duplication (PD), transposed duplication (TRD), and dispersed duplication (DSD) (Supplementary Fig. 3). It showed that PD is the predominant type (27.89%, 1807), followed by WGD (1726, 26.64%) and TD (1190, 18.37%) (Supplementary Fig. 3). All types of duplicated genes within the expanded families were enriched in primary metabolic biosynthesis (Supplementary Fig. 4). Interestingly, genes classified as DSD, TD and PD within the expanded gene families were enriched in secondary metabolic pathways, such as flavonoid biosynthesis, phenylpropanoid biosynthesis and benzoxazinoid biosynthesis (Supplementary Fig. 4), suggesting that dispersed, tandem and proximal duplication played vital roles in broadening the gene repertoire associated with secondary metabolic biosynthesis. Such expansions are indicative of an extensive accumulation of diverse secondary metabolites in *A. dahurica*.

### Accumulation of FCs during root development
To investigate the accumulation pattern of FCs in the root at different sampling dates of *A. dahurica*, we sampled *A. dahurica* roots once a month

from May to October, referred to as S1 (12 weeks after seeds planting), S2 (16 weeks), S3 (20 weeks), S4 (24 weeks), S5 (28 weeks), and S6 (32 weeks), respectively (Fig. 3A). We examined root diameters and content of 17 coumarins, including 14 FCs and 3 upstream simple coumarins, which have important pharmacological effects.

The results revealed that during stages S1 to S5, the diameter of *A. dahurica* roots continued to increase and eventually stabilized at $5.2 \pm 0.1$ cm (Fig. 3B). As the quality control standards of *A. dahurica* in Chinese Pharmacopoeia (2020 Edition), the concentration of imperatorin and isoimperatorin remained consistently high throughout the developmental stages of *A. dahurica* (averaging 96.09 µg/g for imperatorin and 20.40 µg/g for isoimperatorin, respectively) (Fig. 3C). In addition, we found that some upstream substrates and metabolic intermediates involved in the complex FC biosynthesis exhibited distinct changes. For instance, umbelliferone, serving as the entry substance for the biosynthesis of linear and angular FCs, and marmesin, functioning as a substrate in the biosynthesis of linear FCs, showed a decreasing trend in their concentrations with root enlargement. While the concentration of xanthotoxol in *A. dahurica* roots reached the highest in the S5. In summary, despite a slight decline in the overall FCs concentration with root development (Fig. 3D), the continuous increase in root diameter from S1 to S5 resulted in the highest root biomass at S5 (September), which explains the rationale for the common practice in the production of harvesting *A. dahurica* roots before bolting in the autumn season.

### Mining and verification of P8H and P5H

Multi-omics data were integrated to identify candidate CYP450 genes involved in the catalysis of psoralen to xanthotoxol and bergaptol in *A. dahurica*. Based on sequence similarity and conserved domain integrity, we identified 310 *A. dahurica* CYP450 genes, each with >300 amino acids and an FPKM > 0 (Supplementary Data 1,2) in at least one tissue or root stage. Subsequently, a phylogenetic tree was constructed to compare the CYP450 genes with those from *A. thaliana* and those experimentally verified to be involved in FC biosynthesis[1,14,44], resulting in 25 preliminary candidates for P8H and P5H (Fig. 4B). The correlation between metabolite content and gene expression levels in five *A. dahurica* tissues (root, mature leaves, young leaves, stem, and flowers) (Fig. 4A, C) were utilized to further screen candidate genes. Ultimately, based on the correlation between FPKM and metabolite levels in diverse tissues ($r > 0.75$, Supplementary Data 3), we selected two P8H candidates (*AD04G02371* and *AD03G00078*), and four P5H candidates (*AD04G02366*, *AD08G00823*, *AD03G00273* and *AD04G04017*) for downstream experimental validation (Fig. 4B). In the tobacco transient expression system, *AD04G02371*, and *AD043G00078* were found to catalyzes hydroxylation at *C*-8 positions of psoralen into xanthotoxol, while *AD04G02366* react on hydroxylation at *C*-5 positions, significantly enhancing the conversion of psoralen into bergaptol (Fig. 4D). These protein sequences were subsequently named CYP71AZ19, CYP83F95, CYP71AZ18, respectively.

### *CYP71AZ19* results from a *A. dahurica*-specific proximal duplication of *CYP71AZs*

To investigate the evolutionary history of FC biosynthesis pathway in a large phylogenetic framework encompassing the four high-FC angiosperm families, we collected three groups of CYP71 protein sequences: 1) three verified enzymes from *A. dahurica* (CYP83F95, CYP71AZ18 and CYP71AZ19); 2) outgroup representatives, CYP51G1 and CYP71A12 from *A. thaliana*; 3) CYP71 homologs from five Apiaceae species (*A. sinesis*, *L. chuanxiong*, *Peucedanum praeruptorum*, *P. sativa* and *D. carota*), one Rutaceae species (*Citrus limon*), one Maraceae species (*Morus notabilis*) and one Fabaceae species (*Medicago truncatula*). The phylogenetic tree of CYP71s was divided into three clades (Fig. 5A). Notably, genes from Rutaceae, Moraceae, and Fabaceae formed a basal clade (Clade I), distinct from the CYP71AZ and CYP83F subfamilies in Apiaceae. Clade II comprised the CYP71AZ subfamily in the Apiaceae, including CYP71AZ18 and CYP71AZ19; while Clade III encompassed

the CYP83F subfamily in the Apiaceae, including CYP83F95 (Fig. 5A). Interestingly, *M. notabilis* (without FCs) had three homologs, whereas *Ficus carica* (with FCs) lacked any homolog. Together with the observation that homologous genes of the other three families were clustered into a distinct clade from those in Apicaceae, we proposed that the function of CYP71AZ and CYP83F subfamilies was not conserved among high-FC families, and they underwent lineage-specific evolution to acquire the catalytic function in Apiaceae. In contrast, the corresponding step might be catalyzed by enzymes from distinct subfamilies in the other three families.

Based on the synteny analysis, *CYP71AZ18* was collinear with *AS02G00065* in *A. sinensis*, *Pp8G3485* in *P. praeruptorum*, *Ps1G0267* in *P. sativa*, respectively (Fig. 5B). However, collinearity with *CYP71AZ19* was absent in the above-mentioned Apiaceae species (Fig. 5B). Coupled with the finding that *CYP71AZ18* and *CYP71AZ19* are products of proximal duplication (Supplementary Fig. 3), we inferred that *CYP71AZ19* originated from a proximal duplication event of *CYP71AZ18* concomitant with the speciation of *A. dahurica*.

### The landscape of chromatin accessibility in *A. dahurica*

ATAC-seq was applied to investigate the chromatin accessibility of different tissues and during the root developmental stages of *A. dahurica*. By plotting the positions of chromatin accessibility relative to all genes (including coding regions and 3 kb upstream and downstream regions), we found that chromatin accessibility was enriched in transcription start sites (TSS) and transcription end sites (TES) (Fig. 6A, B, Supplementary Fig. 5A, B), showing a similar distribution pattern in leaves and roots at different developmental stages. We next identified 3 833, 33,211, 29,410, 42,281, and 56,543 ACRs in leaves and roots (S0, S1, S3 and S5), respectively. These were assigned to the nearest genes based on the annotation. Among 10 genomic elements (promoter (≤1 kb), promoter (1-2 kb), promoter (2-3 kb), 3'un-translated region (3'UTR), 1st exon (first exon), other exons, 1st intron (first intron), other introns, downstream, distal intergenic regions, 5'UTR), the distal intergenic regions were the most annotated type, which was more abundant in roots (58.03–79.69%) than in leaves (49.84%), followed by the promoter region (≤1 kb) (Fig. 5C). Considering that each genomic element has different proportions on the genome, we further divided the genome into 6 parts, TSS (TSS ± 1 kb), TES (TES ± 1 kb), gene, exon, intron, and intergenic regions, for further enrichment analyses of ACRs, and found that it was mainly enriched in exon, TSS and TES regions, whereas intergenic showed negative enrichment. (Fig. 6D, Supplementary Fig. 6).

We further noticed that genes with ACRs had higher expression levels than genes without ACRs in both leaves and roots (Wilcoxon rank-sum test, $P < 0.0001$) (Fig. 6E, Supplementary Fig. 7). To explore the effects of ACRs in different regions of the genome on gene expression, we categorized ACRs into proximal ACRs (located in 3 kb upstream and 300 bp downstream of the gene) and distal ACRs (located in intergenic regions: > 3 kb upstream of the gene or > 300 bp downstream of the gene). The results showed that genes with both proximal and distal ACRs exhibited the highest expression levels, followed by those possessing only proximal ACRs. Moreover, all three categories of ACR-associated genes showed significantly higher expression levels compared to genes lacking ACRs (Kruskal-Wallis test, $P < 0.0001$) (Fig. 6F).

In addition, we identified differential accessible regions (DARs) between roots and young leaves, with 6494 DARs up-regulated in roots, significantly enriched in the phenylpropanoid biosynthesis pathway (ko00940), and 2381 DARs up-regulated in leaves, significantly enriched in photosynthesis (ko00195) (Supplementary Fig. 8). According to gene expression data, 5620 and 6331 differentially expressed genes (DEGs) were up-regulated in roots and young leaves, respectively. Notably, among these, 1760 root highly expressed genes overlapped with root-specific ACRs, accounting for 27.10% of the root-specific ACRs, and were notably enriched in secondary metabolic pathways, such as phenylpropanoid biosynthesis pathway (ko00940), pertinent to coumarin

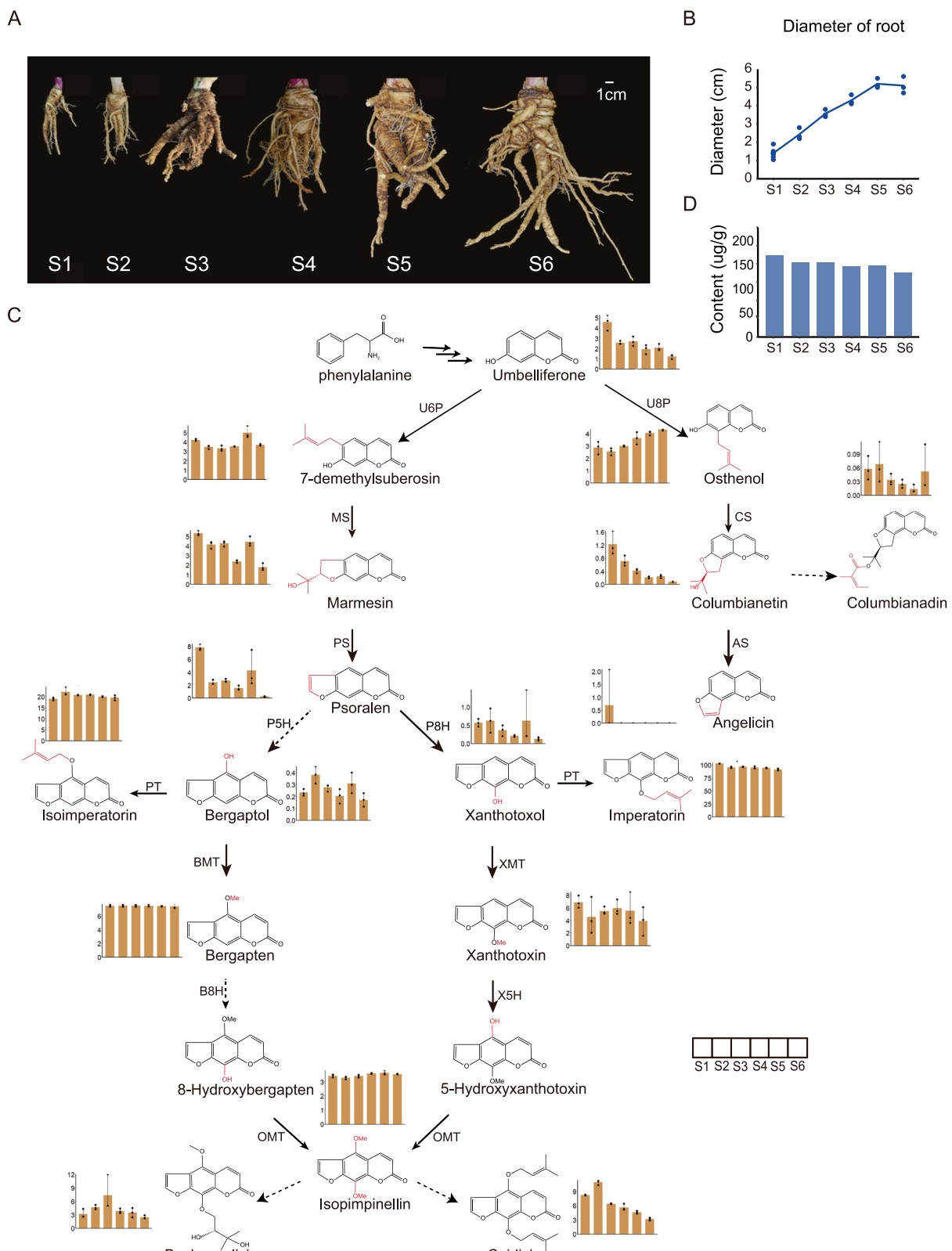

**Fig. 3 | Dynamic profile of root diameters and FCs concentration during root development of *A. dahurica*.** **A** Root phenotypes sampled at six time points from May to October after planting in March. **B** Root diameters. **C** Concentration of 17 coumarins in the biosynthetic pathway and (**D**) overall concentration of 14 FCs throughout *A. dahurica* root development. The bar chart, progressing from left to right (S1-S6), depicts metabolite concentrations in μg/g. The error bars denote the maximum and minimum values.

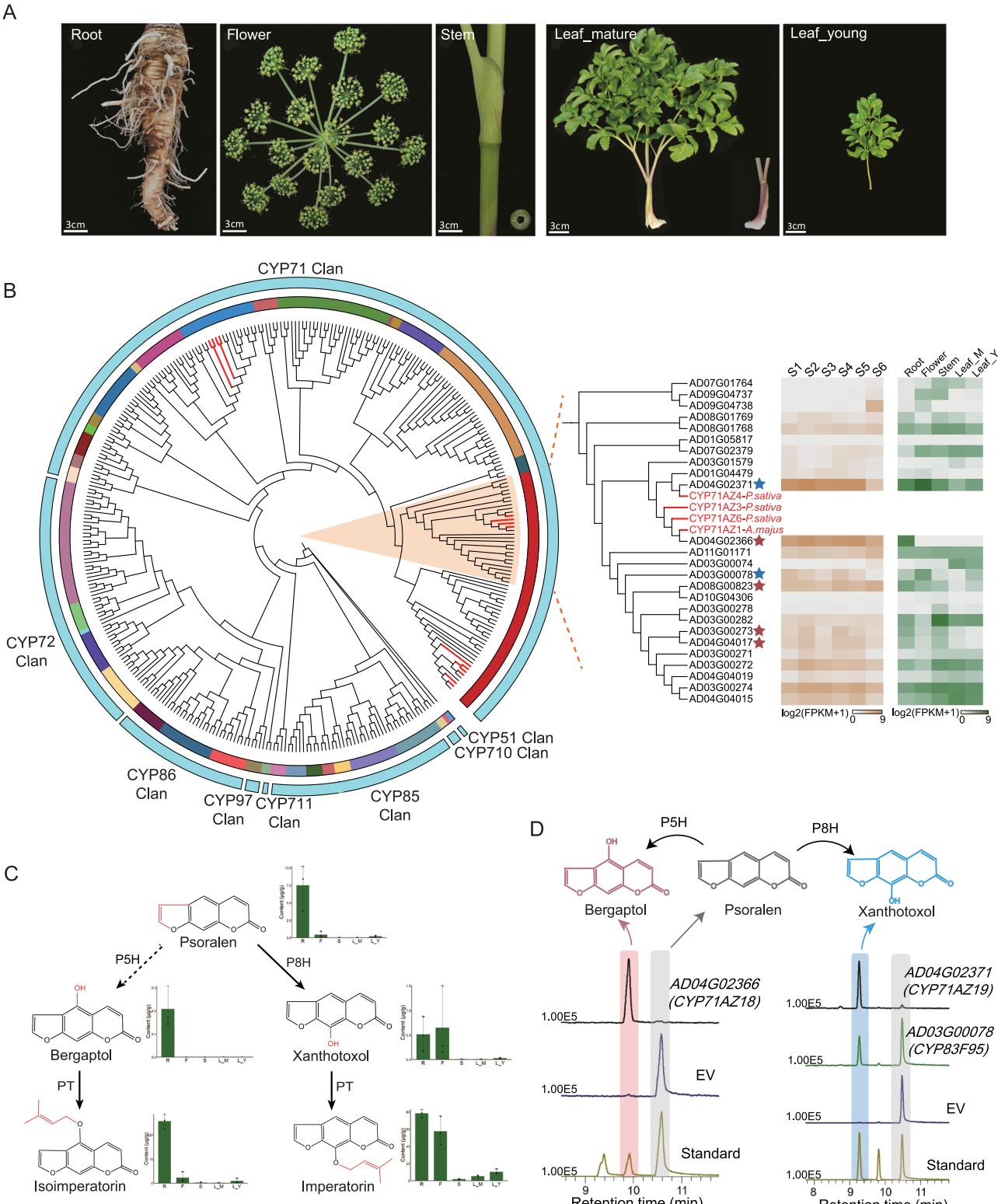

**Fig. 4 | Mining and validation of candidate genes for P5H and P8H. A** Sampling of different tissues of *A. dahurica*. **B** Phylogenetic tree of CYPs from *A. dahurica* for screening candidate genes responsible for FC biosynthetic pathway. The orange shaded lineages represent 25 preliminary candidate genes for P8H and P5H. The red font represents experimentally validated genes involved in furanocoumarin biosynthesis. The heatmap represents gene expression level (FPKM), with blue stars indicating P8H candidate genes and red stars indicating P5H candidate genes.

**C** Concentration of five FCs among five tissues of *A. dahurica*. The error bars denote the maximum and minimum value. **D** LC-MS traces of reactions between psoralen and candidate genes. The left represents the production of bergaptol (the red shaded box) in the indicated combinations of psoralen (gray) and *CYP71AZ18*. The right represents the production of xanthotoxol (blue) in the indicated combinations of psoralen (gray) and *CYP71AZ19* or *CPY83F95*.

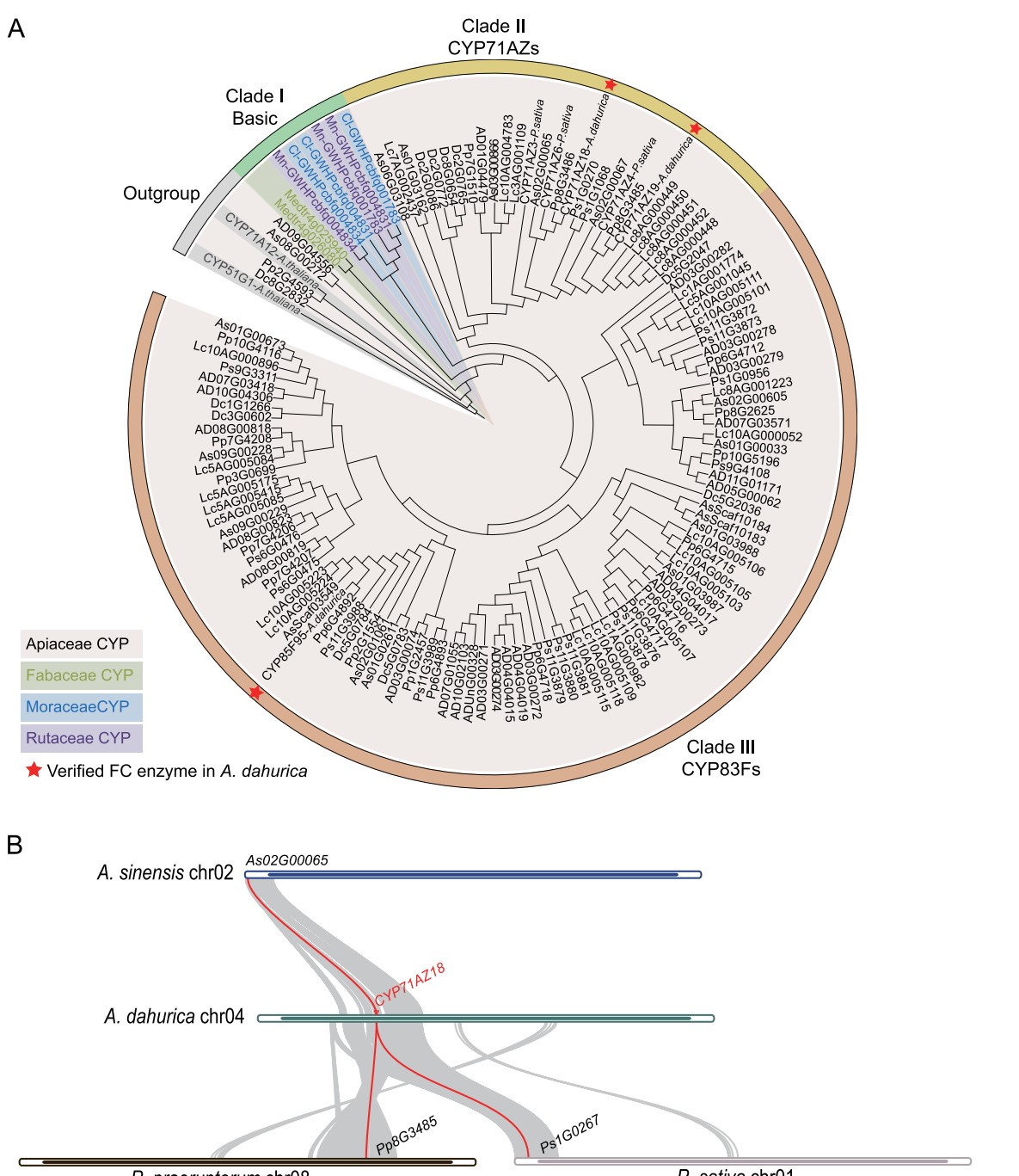

**Fig. 5 | Evolutionary history of the CYP71AZ and CYP83F subfamily in Apiaceae, Rutaceae, Moraceae and Fabaceae. A** Phylogenetic tree of the CYP71s family. **B** The syntenic relationship of *CYP71AZ18* and *CYP71AZ19* with homologous genes in *A.* *sinensis*, *P. praeruptorum* and *P. sativa*. Red lines connect homologs and *CYP71AZ18* between any two syntenic regions.

biosynthesis (Fig. 6G, H). In contrast, only 295 leaf-highly expressed genes overlapped with leaf-specific ACRs, accounting for 12.39% of the leaf-specific ACRs and significantly enriched in the photosynthesis (ko00195) (Fig. 6G, H). The biosynthesis of FC begins with phenylalanine (Fig. 1), which undergoes a series of deamination, hydroxylation, and cyclization reactions to produce umbelliferone, the shared substrate in the FC biosynthetic pathway. Taken together, ACRs play a critical role in regulating gene expression, particularly within the phenylpropanoid biosynthesis pathway in roots, ultimately regulating the accumulation of FCs.

## Chromatin accessibility is associated with FC biosynthesis regulation in *A. dahurica*

To illustrate the possible regulation of chromatin accessibility on the expression of genes related to FC biosynthesis, we further focused on the relationship between gene expression and chromatin accessibility in roots and leaves for the three validated genes. Compared with leaves, *CYP71AZ19* and *CYP71AZ18* showed higher chromatin accessibility and gene expression in roots, whereas *CYP83F95* had higher chromatin accessibility in roots although its expression level was similar in roots and leaves (Fig. 7A). In addition, genes with high chromatin accessibility in roots enriched some TF

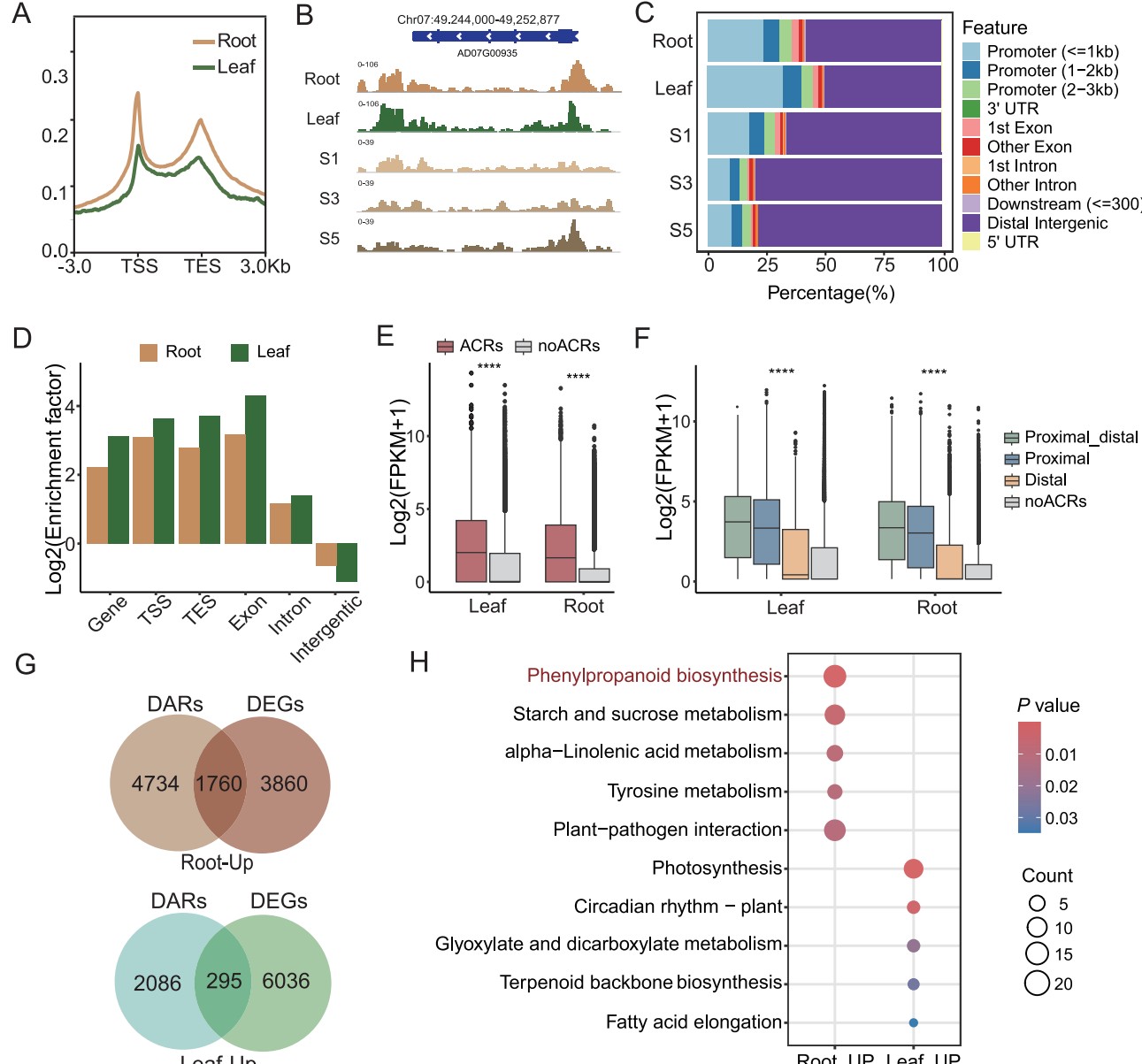

**Fig. 6 | The landscape of chromatin accessibility in *A. dahurica*. A** Chromatin accessibility profiles in gene regions. The 3 kb upstream and downstream flanking coding regions were aligned for all genes. TSS transcription start site, TES transcription end site. **B** Snapshot of genomic regions. **C** Distribution of ATAC-seq peaks within functional genomic elements in *A. dahurica*. **D** Enrichment factor of peaks (ACRs) in gene, TSS (within 1 kb upstream and downstream of the TSS), TES (within 1 kb upstream and downstream of the TES), exon, intron, and intergenic regions. **E** Expression levels of genes with or without related ACRs in root and leaf. **F** Expression levels of genes associated with distal ACRs and proximal ACRs. **G** Venn diagrams show intersection of DARs and DEGs. **H** KEGG enrichment analysis of upregulated ACR-related genes in roots and leaves. Red font represents the KEGG pathway involved in coumarin biosynthesis.

binding motifs associated with stress response or root development, such as RAP2.11 facilitating plant adaptation to abiotic stressors, as well as WRKY aiding in plant defense against pathogen (Fig. 7B, Supplementary Fig. 9), which is consistent with the biodefense function of FCs.

In order to investigate how chromosome accessibility might impact expression of biosynthetic genes in the complete FC biosynthetic pathway, we treated genes exhibiting over 75% homology to queries (genes with experimentally validated functions; Supplementary Data 4) as candidate genes involved in FC biosynthetic pathway, and quantified their expression levels and chromatin accessibility. It revealed that while the expression patterns of these genes in roots and leaves did not consistently mirror the profile of metabolite content, they were consistently correlated with chromatin accessibility (Fig. 7C). Notably, as long as the identified candidate genes exhibited chromatin accessibility, expression levels and downstream

product concentrations would be observed. These genes represented promising candidates for subsequent experimental validation. For instance, among the five candidate genes identified for the conversion of xanthotoxol to imperatorin, *AD01G04637* emerged as a top priority for experimental validation due to its consistent profile in chromatin accessibility, gene expression, and metabolite content distribution.

## Discussion

Here, we constructed a high-quality chromosome-level reference genome of *A. dahurica*. gene family expansion in *A. dahurica* was mainly driven by PD (27.89%), WGD (26.64%), and TD (18.37%); while PD and TD were specifically and significantly enriched in secondary metabolic pathways, including phenylpropanoid biosynthesis (the shared upstream pathway of coumarin biosynthesis) and flavonoid biosynthesis, which were curcial for

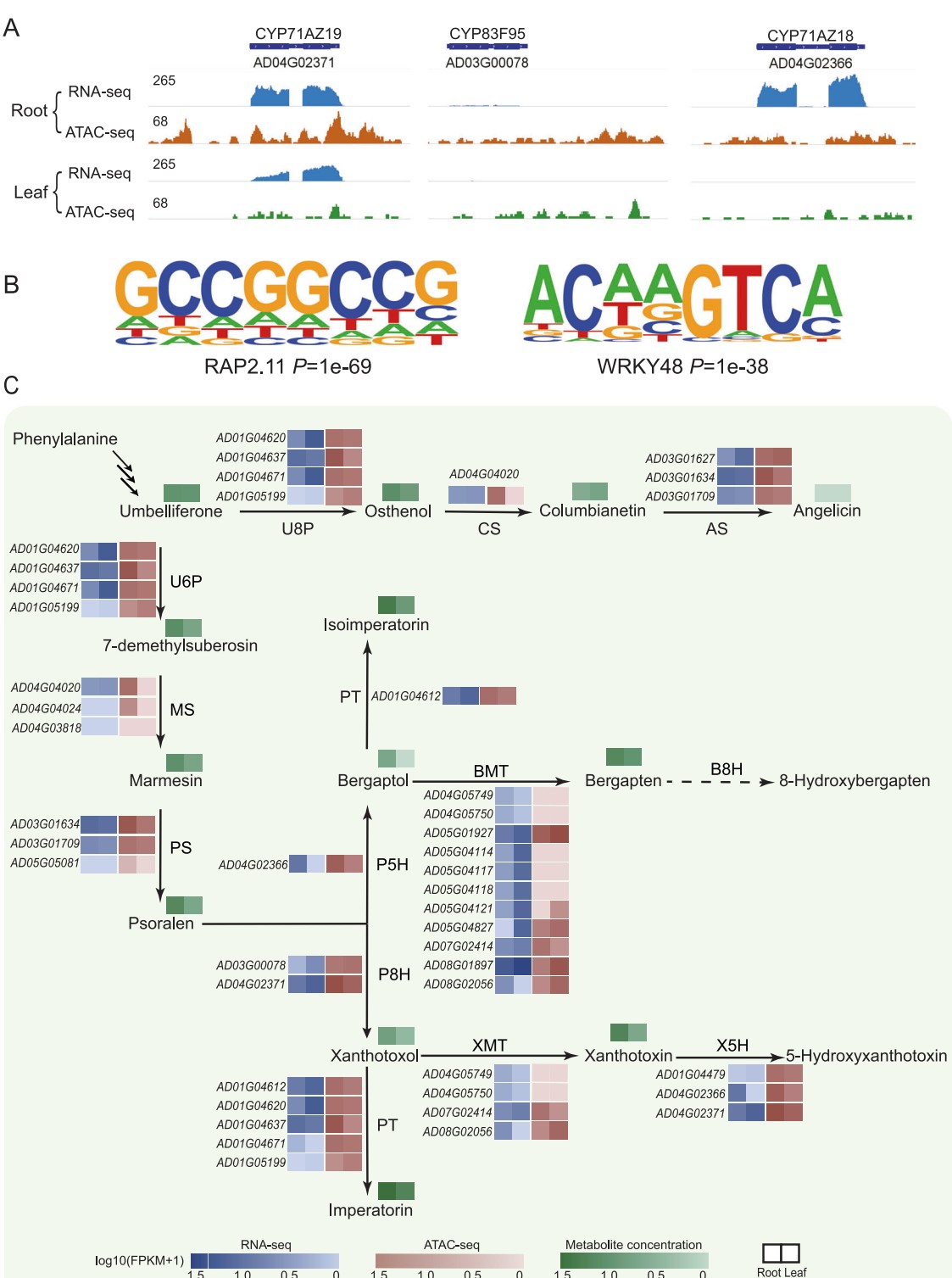

**Fig. 7 | Integrated RNA-seq and ATAC-seq data showing the changes in chromatin accessibility landscape. A** Integrated genome browser view of RNA-seq and ATAC-seq for *CYP71AZ18*, *CYP71AZ19* and *CYP83F95*. **B** Consensus motifs enriched in root specific ACRs. The left demonstrates RAP2.11, and the right exhibits WRKY48. *E*-value generated by HMMER was shown under the motif. **C** Gene expression, chromatin accessibility of candidate genes and metabolite concentration in FC biosynthetic pathway in both roots and leaves of *A. dahurica*. The bule box represents RNA-seq, red denotes ATAC-seq, and green implies metabolite concentration.

environmental adaptation. Bioinformatic analysis, along with in vivo enzymatic assay, confirmed that CYP71AZ18 is involved in the biosynthesis of bergaptol, whereas CYP71AZ19 and CYP83F95 contribute to the biosynthesis of xanthotoxol, elucidated the biosynthetic pathway leading from phenylalanine to imperatorin and isoimperatorin. Notably, *CYP71AZ19* originated from a proximal duplication event of *CYP71AZ18* on the speciation of *A. dahurica*, subsequently undergoing neofunctionalization. The ATAC-seq analysis demonstrated that chromatin accessibility was

positively correlated with gene expression, especially in the proximal ACRs. Notably, the expression of FC biosynthesis genes was also regulated by chromatin accessibility. Furthermore, we investigated the accumulation patterns of 17 coumarins during the root developmental stages of *A. dahurica* and found a gradual decrease in FCs concentration as the roots developed. Our findings provide new insights into the biosynthetic pathway of FC, its evolutionary history, and the epigenetic regulation of secondary metabolite biosynthesis.

Understanding the accumulation pattern of important pharmacologically active compounds in medicinal plants throughout their developmental stages is crucial for optimizing the harvest time and enhancing resource utilization[45]. Our study reveals that the concentration of imperatorin and the 17 coumarins peaked at the S1 period (12 weeks), whereas isoimperatorin reached its maximum level at the S2 period (16 weeks), which is partially consistent with Liang et al.'s observation that the highest concentrations of imperatorin and isoimperatorin peaked at 23 weeks of root growth instead of the harvest time in *A. dahurica*[28]. On the one hand, FCs are a class of defensive metabolites primarily distributed in the phloem of roots rather than the xylem, to enable timely and effective responses to various pathogens and phytophagous insects[46]. Thus, the observed decline may be attributed to root enlargement, which increases the proportion of xylem and causes the proportion of phloem to become relatively contracted, leading to a reduction in coumarin concentration per unit volume. On the other hand, biosynthesis of secondary metabolites demands significant energy expenditure and plants tend to balance energy consumption between growth and defense[47,48]. As juvenile plants mature and their tolerance to pathogens and phytophagous insects increases, the observed decrease in FCs concentration in the roots compared to the seedling stage may reflect a regulatory strategy employed by the plant to optimize resource allocation between growth and defense. However, given that the root diameter of *A. dahurica* continued to increase during the S1-S5 period, with a corresponding increase in root biomass, harvesting during the S5 period remains sensible. The sustained high levels of imperatorin and isoimperatorin observed throughout root development may reflect their pivotal role in biodefense. Given the defensive function of FCs, the application of moderate stress during the developmental stages of *A. dahurica* may promote coumarin accumulation in the roots; however, its potential effects on yield should be carefully evaluated.

The FC biosynthesis pathway represents a compelling case of convergent evolution in higher plants, suggesting that distinct enzyme families may have been independently recruited in different plant families to catalyze the same reactional steps[21,49]. CYP71AZ18 catalyzes hydroxylation at *C*-5 positions of psoralen, while CYP71AZ19 and CYP83F95 react on hydroxylation at *C*-8 positions, contributing to the final diversity of FCs in *A. dahurica* (Fig. 5). The identification of CYP71AZ18 fills the sole missing step in the biosynthesis pathways of imperatorin and isoimperatorin, the two standard compounds of *A. dahurica*[25]. Several reports highlighted that lineage-specific tandem duplications frequently occurred in the CYP71 clan, elucidating that the genetic redundancy ensuing from duplication events serves as a reservoir for functional innovation[50,51]. Our study revealed that *A. dahurica*-specific proximal duplication might drive copy number expansion and functional diversification of CYP71AZ members (Fig. 5), which might enrich the FC diversity and enable adaptation to local environments or defending biotic stress in *A. dahurica*[52]. Previous research on the gradual establishment of the complex coumarin biosynthetic pathway in Apiaceae suggested that late-origin might explain the limited distribution of complex coumarin-containing taxa restricted to derived in Apiaceae[15]. Therefore, additional metabolomic and genetic data are required to completely elucidate origin and evolution of FC biosynthesis pathway in Apiaceae. Moreover, given the convergent evolution of FC biosynthesis and functional convergence observed between CYP71AZ19 and CYP83F95, it is essential to adopt broad thresholds in candidate gene screening based on phylogenetic clades. Lineage-specific evolutionary analyses, transcriptomic assays, biosynthetic gene cluster identifications, and stress response studies, are complementary strategies in candidate gene screening.

Chromatin accessibility is crucial for maintaining precise regulation of gene expression[34,35,53,54]. The regulatory elements controlling gene expression can be broadly categorized into three types: promoters, enhancers, and boundary elements (such as insulators and boundary enhancers), among which, promoters are primarily located at the TSS while enhancers and boundary elements are often situated at distal positions away from genes[55]. The distribution of ACRs in *A. dahurica* is notably enriched in the proximal region of genes, mainly exon, TSS and TES. And these proximal ACRs have a stronger impact on gene expression compared with those distal counterparts, which implies that a potentially dominant role of proximal ACRs in transcriptional regulation through the regulation of promoter sequences and transcriptional accessibility of gene regions, which aligns with previous research[30,56]. Furthermore, genes with both proximal and distal ACRs exhibited the highest expression levels implying a synergistic regulatory interplay between proximal and distal ACRs in modulating gene expression. Moreover, metabolite-enriched roots exhibited a larger number of DARs and DEGs with consistent alteration patterns compared to leaves. Genes associated with up-regulated DARs and up-regulated expression levels in roots were enriched in diverse metabolic pathways, including phenylalanine biosynthesis and phenylalanine metabolism (upstream pathway of coumarin biosynthesis). The three experimentally verified CYP450s displayed a higher chromatin accessibility pattern in roots than leaves, concomitant with the distribution trends of catalytic products, which further corroborates the pivotal role of chromatin accessibility in regulating the synthesis of secondary metabolites in plants. This corroborates the significant influence of chromatin accessibility on secondary metabolite synthesis, as well as plant growth and development. Given that biotic and abiotic stresses can induce the biosynthesis of FC in *A. dahurica*, stress-responding related motifs were enriched among the root up-regulated ACRs. Our study establishes a theoretical foundation for leveraging epigenetic regulation to enhance the biosynthesis of FC. The concentration of FCs is influenced by their biosynthesis, degradation, and transportation, and thus may not fully correlate with the expression levels of genes catalyzing these processes. ATAC-seq data can serve as a complementary tool for screening candidate genes, to enhance screening efficiency and prevent the exclusion of functional genes due to overly stringent criteria.

In conclusion, our study provides a valuable genomic resource of *A. dahurica*, enriches the secondary metabolite profile during its root development, fills an enzyme gap in the FC biosynthetic pathway, explores the evolution of the FC biosynthetic genes, and dissects the epigenetic regulation of gene expression and metabolite content. This research advances our knowledge of the FC biosynthesis pathway and the role of lineage-specific duplication of CYP450 in diversifying FCs and contributes to the understanding of the impact of epigenetic regulation on gene expression, which provides insights into metabolic production of FCs via biosynthetic technology. Additionally, we provide genomic, transcriptomic, metabolomic, and ATAC-seq data, which serve as valuable resources for future studies on the biosynthesis and evolution of enzymes involved in other important metabolites in *A. dahurica*, such as volatile terpenes.

## Methods
### Plant materials
Seeds of *A. dahurica* were procured from its origin production area, Suining City (Sichuan, China), and were subsequently planted in the greenhouse of the Agricultural Genomics Institute at Shenzhen (Chinese Academy of Agricultural Sciences, China). To study the root development process of *A. dahurica*, seeds were planted in March 2021. Sampling commenced in May 2021, with a monthly interval, resulting in a total of six sampling events. Another set of *A. dahurica* plants was planted in August 2020. Upon their flowering in April 2021, five distinct tissues were sampled, including roots, stems, mature leaves, young leaves, and flowers.

*Agrobacterium tumefaciens*-mediated transient expression was performed using 4- to 5-week-old *Nicotiana benthamiana* plants. The seedlings were grown in a glasshouse under lights with a 16 h/8 h light/dark cycle.

## Genome size estimation

Both k-mer and flow cytometry methods were used for *A. dahurica* genome size estimation. Fresh leaves were vertically chopped with a disposable razor blade to release nuclei in cold LB01 lysis buffer (15 mmol/L Tris, 2 mmol/L Na2EDTA, 0.5 mmol/L spermine tetrahydrochloride, 80 mmol/L KCl, 20 mmol/L NaCl, 0.1% (v/v) TritonX-100, 15 mmol/L β-mercaptoethano, pH 7.0–8.0). The nuclei suspension was filtered through a 40 μm cell strainer, stained with 20 μg mL-1 propidium iodide (DNA fluorochrome; Thermo Fisher Scientific, MA, USA) and 20 μg mL-1 RNase A (Thermo Fisher Scientific, MA, USA), and ice-bathed for 30 min in the dark. The fluorescence intensity of stained nuclei was analyzed with CytoFLEX (Beckman Coulter, FL, USA). The value of nuclear DNA was calculated by comparing the nuclear peaks on a linear scale with the peak for *Foeniculum vulgare* Mill. (1 C = 1.01 G) and *A. sinensis* (1 C = 2.37 G) using CytExpert v.2.3 (Beckman Coulter, IN, USA). Clean whole genome sequence (WGS) reads were analyzed to obtain the 19-mer distribution with Jellyfish v2.3.0[57]. The output file was used as the input for GenomeScope 1.0[58] to estimate the genome size and heterozygosity rate.

## DNA and RNA preparation and sequencing

Fresh leaves of *A. dahurica* were harvested for the extraction of high molecular weight (HMW) genomic DNA with the DNeasy Plant Mini Kit (Qiagen, Hilden, Germany). A quantity exceeding 50 μg of HMW DNA was used for the construction of SMRTbell™ libraries, which were prepared using the PacBio SMRTbell Express Template Prep Kit 2.0 (Pacific Biosciences, CA, USA). These libraries were sequenced on the PacBio Sequel II platform (Pacific Biosciences, CA, USA) with the circular consensus sequencing (CCS) mode.

For the Hi-C approach, plant tissues underwent formaldehyde treatment for fixation, with the cross-linked DNA subsequently digested by DpnII throughout the night. The digested fragments' sticky ends were then biotinylated and ligated in a random fashion. These chimeric fragments, reflecting the original cross-linked physical interactions, were further processed into paired-end sequencing libraries. The sequencing phase for these libraries was conducted on the Illumina NovaSeq 6000 platform (Illumina, CA, USA), generating 2 × 150 bp reads.

Concurrently, total RNA was extracted from the roots across seven different growth periods, in addition to flowers, young leaves, old leaves, and stems, employing the RNAprep Pure Plant Kit (TIANGEN, Beijing, China). cDNA synthesis was carried out with 20 μg total RNA, Rever Tra Ace (TOYOBO, Osaka, Japan) and oligo (dT) primers, adhering to the instructions of the user manual. Libraries were prepared using the NEB Ultra II RNA Library Prep Kit (New England Biolabs, MA, USA) and then sequenced on the Illumina NovaSeq 6000 platform (Illumina, CA, USA).

## Genome assembly and annotation

The genome assembly of *A. dahurica* was achieved by combining data from PacBio CCS and Hi-C technologies, employing Hifiasm v0.15.5-r350 software with default parameters (Supplementary Table 1)[59]. Subsequently, Purge Haplotigs v1.1.1[60] was used to remove redundant sequences because of a very high heterozygosity observed in *A. dahurica* genome. Next Quality-controlled Hi-C reads were then aligned to the contig assembly of *A. dahurica* with Juicer v1.6[61]. A preliminary chromosome-level assembly was generated automatically with the 3D-DNA v180114 pipeline to rectify wrong assemblies, orders and orientations[62]. Further refinement and validation of this assembly were conducted manually through the Juicebox Assembly Tools v2.18 (https://github.com/aidenlab/Juicebox), a crucial step in enhancing the accuracy of assembly. Finally, the quality of the chromosome-level genome assembly was rigorously assessed with Benchmarking Universal Single-Copy Orthologs (BUSCO v5.1.2) and LTR Assembly Index (LAI)[63,64].

We employed the EDTA v2.0.1[65] for the comprehensive identification of transposable elements, encompassing LTR, TIR, and non-TIR elements. To predict coding gene structures within the repeat-masked genome, we utilized a multifaceted approach that included ab initio predictions,

evidence from homologous proteins, and transcriptome data of 11 tissues, including roots from seven developmental stages, stems, leaves, and flowers were sequenced (Supplementary Table 2). For the de novo prediction of protein-coding genes, we used AUGUSTUS v.2.3.3[66]. MAKER v3.01.03 pipeline[67] was used to annotate gene structures in our study and protein sequences from species of *D. carota* and *L. chuanxiong*. Further annotations of protein-coding genes were conducted by GFAP[68] and EGGNOG-MAPPER v.1.0.3[69] to the KEGG, GO (http://geneontology.org) and PFAM databases.

## Phylogenetic analyses

Paralogs and orthologs were identified among seven plant species: *C. asiatica*[70], *B. chinense*[71], *D. carota*[72], *A. dahurica*, *A. sinensis*[4] and *L. chuanxiong* haplotype A[41] with the OrthoFinder v2.5.2[73], and protein sequences of single-copy orthologous genes were used to construct a phylogenetic tree. The concatenated amino acid sequences were aligned using MAFFT v7.271[74] and trimmed with trimAI v1.4.rev22[75]. A maximum likelihood phylogenetic tree was constructed using RAxML v.8.2.12 of a PROT-GAMMAJTT model with 1000 bootstrap replicates[76], and *C. asiatica* was used as the outgroup. The species tree was then used as an input to estimate divergence time in the MCMCTree program of the PAML package[77]. Fossil time of divergence between *B. chinense* and *D. carota* was used for time calibrations from TIMETREE. The expansion and contraction of gene families were inferred with CAFE5 v5.0[78] based on the chronogram of the above-mentioned nine plant species.

Syntenic analyses and gene duplication identification Syntenic blocks within one species or between two species were defined by MCscanX v1.1.11[79] based on homologous gene sets using BLASTP v2.10.0 (E-value < 1e-5; the number of genes required to call a syntenic block ≥ 5). To further identify the pattern of genome-wide duplications in *A. dahurica*, duplicated genes were divided into five categories: whole-genome duplication (WGD), tandem duplication (TD), proximal duplication (PD), transposed duplication (TRD), dispersed duplication (DSD), using DupGen_Finder v1.12[80] with the default parameters. Genes in the five duplicate categories were further fed with KEGG terms enrichment analysis with TBtools v2.030[81].

## Transcriptome data analyses

RNA-Seq reads of *A. dahurica* was assembled in a reference genome-based strategy with Hisat2 v2.2.1[82]. Gene expression was normalized as fragments per kilobase of exon model per million reads mapped (FPKM) with Stringtie v2.1.4[83]. Differentially expressed genes (DEGs) were selected using DESeq2 v2_1.36.0[84] when $|\log_2 \text{FoldChange(FC)}| > 1$ with adjusted *P*-value ($P_{\text{adj}}$) < 0.05. All genes were used for both GO enrichment and KEGG pathway analysis with eggNOG-mapper v2.1.9 (http://eggnog-mapper.embl.de/)[69]. The R package clusterProfiler v4.4.4[85] was used to perform functional and pathway enrichment analysis, with *P* < 0.05 as a criterion for selecting enriched items.

## Multi-omics mining for candidate CYP450 genes

To identify potential CYP450 genes in *A. dahurica*, we integrated three distinct approaches to preliminarily obtain the CYP450 gene set. 1) We conducted a search based on conserved domains (Pfam ID: PF00067) using HMMER v3.3[86]; 2) we utilized a sequence similarity approach, employing the protein sequence of the CYP450 enzyme CYP71AZ4, previously characterized in *P. sativa* (Apiaceae), as a query sequence for local BLAST in the *A. dahurica* genome (--evalue 1e-6); 3) we excluded potential pseudogenes (predicted protein sequences < 300 amino acids or FPKM = 0 in every sample). We considered the intersecting sequences obtained from three methods as putative CYP450 genes in *A. dahurica*.

To narrow down the set of candidate genes, a phylogenetic analysis of these sequences with CYP450 protein sequences in *A. thaliana*, as well as sequences verified to be involved in furanocoumarin synthesis. All sequences were aligned by MAFFT v7.487[87]. Phylogenetic reconstructions were implemented in IQtree v2.2.5[88] to infer the maximum-likelihood (ML) tree.

The final tree was visualized and annotated in iTOL v6.7.6 (https://itol.embl.de/)[89]. With reference to CYP450 sequences of *A. thaliana*, we categorized the phylogenetic tree into seven clades: CYP71 Clan, CYP86 Clan, CYP97 Clan, CYP72 Clan, CYP711 Clan, CYP85 Clan, and CYP51 Clan. Based on the phylogenetic relationships, we initially screened the 25 *A. dahurica* genes located in CYP71 Clan and clustered with CYP71AZ were identified as preliminary candidates for P8H and P5H (Fig. 4). We further calculated the Pearson correlation coefficients between gene expression levels and secondary metabolite contents from multiple tissues and root developmental stages of the 25 preliminary candidates in *A. dahurica*. Ultimately, we obtained two genes strongly correlated with xanthotoxol content ($r > 0.75$) and four genes correlated with bergaptol content, which were subsequently prioritized for experimental validation.

### Functional characterization for candidate CYPs in *N. benthamiana*

Two psoralen-8-hydroxylases along with four psoralen-5-hydroxylases were cloned into the pSuper1300 vector and subsequently introduced into the GV3101 strains of *Agrobacterium* electrocompetent cells. The GV3101 strains underwent 16 h cultivation in Luria-Bertani (LB) medium supplemented with kanamycin (50 μg/mL) at 28 °C and 200 rpm. Thereafter, 500 μL of the GV3101 suspension was inoculated into 30 mL of fresh LB medium containing rifampicin (25 μg/mL) and kanamycin (50 μg/mL), followed by overnight incubation. After centrifugation at 4000 rpm for 10 min, GV3101 cells were collected and resuspended in MMA buffer (comprising 100 μM acetosyringone, 10 mM MES, and 10 mM MgCl₂, pH 5.6) for a duration of 2–3 h at ambient temperature to optical density at 600 nm (OD600) approximately 0.5-0.6. The strain was subsequently introduced into the abaxial surface of three leaves sourced from disparate 4- to 5-week-old *N. benthamiana* seedlings utilizing a needleless syringe (1 mL). After *N. benthamiana* was dark-treated for 12 h and grown under normal conditions for 3 days, psoralen at a concentration of 0.1 mM was introduced into the leaves infiltrated with *Agrobacterium* to enzymatic activity assessment.

### Identification of metabolites

Due to the photosensitivity of coumarin, all sample solutions were prepared under shaded conditions. After homogenizing the frozen tissue samples into a fine powder on a ball mill, approximately 1 g of sample powder was weighed and added to 5 mL of ethyl acetate. The mixture was vortexed for 1 min, sonicated for 30 min, and then centrifuged at 4 °C and 13000 g for 15 min to collect the supernatant, which was filtered through a 0.22 μm membrane and suspended dry at −70 °C in a refrigerated vacuum centrifugal concentrator. The resulting solid residue was re-solubilized with 1 mL methanol and filtered again. 1 mL extracts were used for Liquid Chromatography-Mass Spectrometry (LC-MS) analysis using Thermo Scientific Vanquish Ultra-High Performance Liquid Chromatography System and Thermo Scientific TSQ Quantum Access Max Triple Quadrupole Mass Spectrometer (both from Thermo-Fisher Scientific) equipped with an ACQUITY UPLC®BEH C18 column (Waters Ltd., MA, USA) (2.1×100 mm, 2.5 μm). Acetonitrile was used as mobile phase A and water with 0.1% formic acid was used as mobile phase B, and the flow rate was 0.3 mL/min. Metabolite identification was performed using authentic standards purchased from suppliers.

### Phylogenetic analysis of the CYP71 family

The construction of a phylogenetic tree incorporating both previously published and newly identified CYP protein sequences was a pivotal step toward understanding their evolutionary relationships. Initially, experimentally verified CYP71AZ18, CYP71AZ19 and CYP83F95 were used as query to Blastn against *D. carota*, *L. chuanxiong* haplotype A and *A. sinensis*, *P. praeruptorum*, *P, sativa*, *C. limon*, *M. notabilis* and *M. truncatula* with E-value < 1e-6, sequence identity > 55% and amino acid length > 100 as a cutoff. CYP51G1 and CYP71A12 in *A. thaliana* were designated as outgroups. Protein sequences of these CYP genes were then aligned using MAFFT

v7.271[74]. Maximum likelihood trees were constructed with IQ-TREE v2.1.4[90]. The final phylogenetic trees were annotated and depicted with iTOL v6[91]. Syntenic analysis was performed by One Step MCScanX of TBtools v2.154[81].

### ATAC-seq library construction, sequencing and analysis

ATAC-seq was conducted following a previously established protocol[92]. Briefly, 1 g of frozen sample was minced in 1 mL of ice lysis buffer (15 mM Tris-HCl pH 7.5, 20 mM NaCl, 80 mM KCl, 0.5 mM spermine, 5 mM 2-Mercaptoethanol, 0.2% TritonX-100). The resulting slurry containing the nuclei extract was filtered twice through a 40 μm filter. The crude nuclei containing DAPI (Sigma, MO, USA) were then loaded onto a flow cytometer (BD FACSCanto, CA, USA) for selection. After centrifugation, the nuclei pellet was washed with Tris-Mg buffer (10 mM Tris-HCl pH 8.0, 5 mM MgCl2), and Tn5 transposomes in 40 μl TTBL buffer (Vazyme, Nanjing, China) were added for a 30 min incubation at 37 °C. Following incubation, the integration products were purified using the NEB Monarch™ DNA Cleanup Kit (New England Biolabs, MA, USA), and library amplification was performed using the NEB Next Ultra II Q5 master mix (New England Biolabs, MA, USA). Finally, the amplified libraries were purified using Hieff NGS® DNA Selection Beads (Yeasen Biotechnology, Shanghai, China).

Raw reads were trimmed with Fastp v0.12.4[93] and then aligned to the *A. dahurica* reference genome with Bowtie2 v2.4.2[94]. PCR duplicates, which may have arisen from amplification, were discarded with Sambamba v0.8.0[95] and low-quality mappings (Q-value < 30) in the resulting BAM files were removed with SAMtools v1.13[96]. Bigwig files were generated by deepTools v3.5.2[97] and then imported into Integrative Genomics Viewer (IGV) v2.17.0[98] to visualize the accessible chromatin landscape. ATAC peak calling was performed with MACS2 v2.2.6[99] using '-g 4.9e9 --keep-dup all --nomodel --shift -100 --extsize 200 -n Leaf'. CHIPSEEKER v1.16.1[100] was used to retrieve the nearest genes around the peak and annotate the genomic region of the peak. The differential accessible regions (DARs) between samples were identified with MACS2 bdgdiff.

### Statistics and reproducibility

Statistical analyses were performed in R (v4.3.1). Wilcoxon rank-sum test was applied to compare two-group comparisons and Kruskal-Wallis test for multi-group analyses. For each tissue sample, we prepared three technical replicates of RNA-seq libraries, three technical replicates of metabolite detection, and one or two technical replicates of ATAC-seq libraries. Each replicate consisted of tissues pooled from three to seven plants.

### Reporting summary

Further information on research design is available in the Nature Portfolio Reporting Summary linked to this article.

### Data availability

All data needed to evaluate the conclusions in the paper are present in the paper and/or the Supplementary Data. The raw sequence data reported in this study, including the PacBio HiFi reads, Hi-C reads, RNA-seq and ATAC-seq data, have been deposited in the Genome Sequence Archive in National Genomics Data Center, China National Center for Bioinformation / Beijing Institute of Genomics, Chinese Academy of Sciences (GSA: CRA024200) that are publicly accessible at https://ngdc.cncb.ac.cn/gsa. The assembled genome files are available at figshare (https://figshare.com/s/bea35667a2b27179a121). The source data behind the graphs in the paper can be found in Supplementary Data 5.

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

## Acknowledgements

We thank the support from Innovation Program of Chinese Academy of Agricultural Sciences. We also thank D. Nelson of P450 nomenclature committee for naming the CYPs. This work was supported by the National Key Research and Development Program of China, grant 2023YFA0915800; National Natural Science Foundation of China, grants 32300223, 32070242, and 82373837; Shenzhen Fundamental Research Program, grant 20220817165436004; Shenzhen Science and Technology Program, grant KQTD2016113010482651; Key Project at Central Government Level (The ability establishment of sustainable use for valuable Chinese medicine resources), grant 2060302; Special Funds for Science Technology Innovation and Industrial Development of Shenzhen Dapeng New District, grants RC201901-05 and PT201901-19; Basic and Applied Basic Research Fund of Guangdong, grant 2020A1515110912; Science, Technology, and Innovation Commission of Shenzhen Municipality of China, grant ZDSYS20200811142605017.

## Author contributions

L.W. conceived and designed the study. J.J. prepared the materials. X.H. assembled and annotated the genome, and conducted WGD analysis. Y.L. and Z.L. performed the ATAC experiment. J.J. performed ATAC analysis, screened candidate genes and visualization. Z.L. and D.H. cloned and characterized the candidate genes. J.J., L.L. and X.H. wrote the manuscript. L.W., Z.L., S.S., Y.R. and G.M. revised the manuscript. All authors read and approved the final manuscript. J.J. and X.H. contributed equally to this work.

## Competing interests

The authors declare no competing interests.
