## [Transparent Peer Review file · Communications Biology]

Integrative multi-omics data provide insights into the biosynthesis of furanocoumarins and mechanisms regulating their accumulation in *Angelica dahurica*

Corresponding Author: Professor Li Wang

Version 0:

Reviewer comments:

Reviewer #1

(Remarks to the Author)

Thank you for submitting your manuscript entitled "Integrative multi-omics data elucidating the biosynthesis and regulatory mechanisms of furanocoumarins in *Angelica dahurica*" to Communications biology. The paper mainly constructed a chromosome-level genome for *A. dahurica* and quantified the concentration dynamics of 17 coumarins across six developmental stages of the root. The identification of specific CYP450 enzymes involved in bergaptol and xanthotoxol biosynthesis, along with the evolutionary history of these enzymes, offers valuable information for enhancing the insights into furanocoumarins biosynthetic pathway and the epigenetic regulation of metabolite biosynthesis. The manuscript was rich and reliable in content. I suggest this manuscript can be accepted after the following minor revisions. More exhaustive comments are as follows.

In lines 138-140. "For genome annotation, transcriptomes of 11 tissues, including roots from seven developmental stages, stems, leaves, and flowers were sequenced. By combining *ab initio*, homology-based and transcriptome-based approaches". This part should be moved to materials and methods more appropriately.

In line 154. "*D. carota*, *L. chuanxiong* and *A. sinensis*." The Latin name of the species should be fully listed with its genus when it first appears in the text.

In lines 161-162. "The number of expanded gene families in *A. dahurica* was almost twice as that in *D. carota*, *A. sinensis* and *L. chuanxiong*." How do the authors interpret the implications of having nearly twice as many expanded gene families in *A. dahurica* compared to other species? What are the possible reasons for the difference? Discussion on potential adaptive advantages of these expansions would enhance the understanding. This is just a suggestion.

In line 194. The findings indicate a decline in overall FC concentration despite increased root biomass. What are the potential biological implications of this observation? How are these substrate concentration changes related to FCs accumulation? Discussing this could provide a deeper insight into the metabolic strategy of *A. dahurica* during root development.

In line 200. The manuscript mentions the integration of multi-omics data but lacks details about how different omics layers were combined. It is suggested to clarify their approach to integrating transcriptomic, metabolomic, and any other omics data in the methods section. The correlation analysis used to connect gene expression levels and metabolite content is significant, but the statistical software used to assess this correlation was not specified in the methods section.

In line 212. All gene names should be italic. Check and modify the rest of the manuscript.

In lines 216-217. The authors report that AdP8H1, AdP8H2, and AdP5H1 exhibited catalytic activities in a tobacco transient expression system, but the function of these genes in this pathway is not specifically elucidated here. It is suggested that the specific roles of these enzymes in the metabolism of xanthotoxol and bergaptol should be elaborated.

In lines 219, 222, and 225. "*A.dahurica-specific*" should be changed to "*A. dahurica-specific*". "*A.dahurica genes*" should be

changed to "A. dahurica genes". "(3) CYP71As and CYP71Bs" should be changed to "3) CYP71As and CYP71Bs". Check and modify the rest of the manuscript.

In line 232. This notes that both CYP71AZ and CYP83F subfamilies underwent lineage-specific duplication events in Apiaceae. How might they contribute to the functional diversity of these gene families within the Apiaceae? The analysis states that there is no observed collinearity for CYP71AZ19 with the other genes in Apiaceae species. What are the possible reasons for this absence of collinearity?

In line 289. The conclusions highlight the importance of ACRs in regulating gene expressions, particularly in phenylpropanoid biosynthesis. It is suggested that a brief overview of the roles of key genes within these pathways to enhance the reader's understanding of the biological implications?

In line 309. The findings indicate tissue-specific expression patterns of genes involved in FC biosynthesis in *A. dahurica*. Differences between gene expression patterns and metabolite content were noted. What is the possible reason for this observation?

In lines 494, 503. "*Daucus carota*" and "*B. chinense* and *D. carota*". The species names should be in italics. Check and modify the rest of the manuscript.

In lines 505, 508. CAFE5 and MCscanX. Please add the software version information.

In line 515. "Kyoto Encyclopedia of Genes and Genomes (KEGG)". As already mentioned, it can be abbreviated.

In line 597. "and *Angelica sinensis*". "and" should be in body, not italics. Check and modify the rest of the manuscript.

There are many mistakes in the references. Please ensure the accuracy and completeness of the references. For example: In lines 657, 728. "Nat Methods" and "Plant J". Please check the format of the journal, not abbreviations. Check and modify the rest of the manuscript.

In line 694. "*Aedes aegypti*" should be changed to "*Aedes aegypti*". "*Coptis chinensis*" in Line 739, and "*k*-mers" in Line 805, etc. Please check the article title.

In line 694. "Schulz, A.J.P.o.t.N.A.o.S." The author information is incorrect. The literature lacks a journal title. Please check and modify it.

Some punctuation mistakes should be checked carefully, such as "The family of furocoumarins:." should be "The family of furocoumarins." in Line 757, etc.

In line 991. "Ethics declarations". There is a lack of content here

In line 999. "Supplemental information titles and legends". Many species name format errors, please check carefully.

The "5' UTR" positions overlap in Fig. 5C, please adjust it.

Reviewer #2

(Remarks to the Author)

This paper presents results on the biosynthesis of furanocoumarins in Apiaceae plants. These molecules constitute an interesting subject of evolutionary escalation between plants and insects as has been long ago shown by the groups of May Berenbaum and Mary Schuler. The data were obtained from *Angelica dahurica*, an important plant in traditional Chinese medicine. The main results are (1) the identification of CYP71AZ18, a gene encoding psoralen 5-hydroxylase, a missing step in the synthesis of 5-hydroxy psoralen derivatives, (2) the evolutionary origin of CYP71AZ19, a P450 carrying psoralen 8-hydroxylase activity, from a duplication/neofunctionalization event of CYP71AZ18, (3) the correlation of chromatin accessibility with the expression of genes involved in the synthesis of furanocoumarins and the concentrations of these compounds found in different plant tissues.

Considering that many P450s have now been described in the furanocoumarin pathway and that neofunctionalized duplication has already been demonstrated in furanocoumarins produced by Apiaceae plants, the main novelty of this manuscript lies in (3). The authors have produced a large quantity of data, from genome sequencing of the plant to transcriptional analyses, from characterisation of novel orthologs and paralogs of the CYP71AZ subfamily to furanocoumarin concentrations assessed at different periods of the growth cycle. My main concern is that the manuscript just scratches the surface of different topics that could have been more studied in more depth with the same dataset. I personally expected a more detailed study of one of the 3 aforementioned topics.

Remarks for improving the manuscript as it is:

- L43: the term "charm" is certainly inappropriate here

- L47-52: description of the interest of furanocoumarins in medicine and dermatology is outdated. Since the early 90s, these compounds have been banned from cosmetic products because they are highly carcinogenic, and directly correlated with the rate of malignant melanoma in humans. Psoriasis is now treated with new small molecules such as Deucravacitinib or anti-IL17/23 antibodies for the most severe cases.

- L56: the two proposed references (72 and 74) relating to the biosynthetic genes described to date are confusing. Indeed, none of them correspond to articles describing new biosynthetic enzymes. 72 concerns the metabolic engineering of the furanocoumarin pathway and 74 is a book chapter referring to the diversity of furanocoumarins found in the plant kingdom.

Instead, cite here original work on the discovery of biosynthetic enzymes.

- L62-71: rather than citing 4 references at the end (36, 41, 44, 84), it is better to insert them where they are relevant in this paragraph.
- L106: spelling error for imperatorin and isoimperatorin.
- L130: avoid judging the anchoring rate. It's not impressive, it's just the way it is.
- L138 I'm not convinced by the term "developmental stage" which should imply a different physiological stage. Here it's more about "sampling dates" not necessarily related to ontogeny. Also check the number of sampling dates because I count 6 and not 7 in Fig. 2.
- L154: please write the full genus name of each plant species. I didn't know what *L. chuanxiong* was before going to Fig. 1.
- L195: the decrease in furanocoumarin concentration in late harvested roots could be due to a dilutive effect related to the increase in biomass while furanocoumarin synthesis could have stopped at this stage. Just my opinion...
- L212-217: this paragraph is just too straight to the point. First, the selected genes are only candidates, not P8H or P5H as stated here. Also, please use the international codification for genes (script letters) and protein sequences (capital letters). The real status of these sequences should be clearly written here (orf? cDNA? etc.).
- Part on chromatin accessibility (L291-303). This part is quite confusing because Fig. 6 presents a lot of data that is not commented by the authors. Why comment only on Fig. 6A? After reading this part, it is not clear whether chromatin accessibility is positively correlated or not with transcriptional data of biosynthetic genes. Fig. 6C contains many cDNA sequences presented under each enzymatic step but it is not clear whether these activities have been experimentally assigned or are just hypotheses based on sequence similarities.
- L299-303: There is probably much more to discover in the genomic data regarding transcription factors than just the abscisic acid and WRKY factors mentioned here. This question of transcriptional regulation of the furanocoumarin pathway is completely unresolved to date.
- General comment on the figures: some histograms are too small. The name of the furanocoumarin standards should be added in Fig.3D.

Reviewer #3

(Remarks to the Author)

Integrative multi-omics data elucidating the biosynthesis and regulatory mechanisms of furanocoumarins in *Angelica dahurica* by Ji et al., reported the assembly of a chromosome-level genome for a famous perennial Chinese traditional medicinal plant *Angelica dahurica* and explored the dynamics of 17 coumarins across six developmental stages of roots, which are usually used in clinical treatments, as they are enriched with bioactive volatile terpenoids and coumarins.

The authors in a genome research group have published several genome assemblies of same plant family. All of them contain high levels of volatile terpenoids and coumarins, that might be responsible for their pharmacological activity and supporting their clinical applications. In this *A. dahurica* genome, the authors particularly focused on furanocoumarins (FCs) biosynthesis pathway and structural genes. They proposed CYP71AZ18 was involved in the biosynthesis of bergapton, whereas CYP71AZ19 and CYP83F95 contributed to the biosynthesis of xanthotoxol, with several pieces of biochemical evidence. and also analyzed the evolutionary relationships among these P450 genes, in *Angelica* genus, etc.

While the manuscript about the genome sequence analysis and assembly quality are routinely well done, also proposed genes from researching the genome and transcriptomic databases in the constructed biosynthesis pathways, I see some weak points:

1. The authors may want to check carefully on the characterization of CYP71AZ19/18 and CYP83F95, for reactions and LC-MS examination of the end products in the reactions
2. By comparison of these genes in other *Angelica* species that have been sequenced, how they are different from evolutionary genomic perspective? The enriched FC in *A. dahurica* could be reflected from the gene expansion of duplication of key genes, as compared with *A. sinensis*, *D. carota*, *L. chuanxiong* in Apiaceae family.
3. Or from four plant families: Apiaceae, Rutaceae, Moraceae, and Fabaceae that contain higher level of FC, their indeed divergent or convergent mechanisms are? from the perspective of the P450 gene evolution.
4. As enriched in special volatile terpenoids, *A. dahurica* genome should have footprints of the special evolutionary trajectory of related pathways and gene evolution, which is worthy of further digging of the genome data and transcriptome data for more indications or markers.

Version 1:

Reviewer comments:

Reviewer #1

(Remarks to the Author)

The authors have addressed all my concerns. I am pleased to recommend this excellent work for prompt publication in its current form.

Reviewer #2

(Remarks to the Author)

The authors have made extensive corrections, as requested by the 3 reviewers. The new version of the manuscript reads well and is ready for publication provided that a few typos are corrected:

L361: changed insectsthis by insects

L952: missing final n in Shenzhe?

Reviewer #3

(Remarks to the Author)

The revised manuscript has been appropriately improved, and the authors addressed my concerns. Now, I think it is a nice work with an in-depth demonstration of biosynthesis, regulation, and evolutionary mechanisms for furanocoumarins in *Angelica dahurica*, in comparison with other *Angelica* lineages.

The work now presents significant advances in understanding of FC biosynthesis and evolution based on multi-omics data, also containing solid biochemical evidence for CYP71AZ18-catalyzed hydroxylation at C-5 position of psoralen and CYP71AZ19- and CYP83F95-catalyzed hydroxylation at C-8 position to produce xanthotoxol.

Response to reviewers

Dear Editor,

Thank you for considering our manuscript, “Integrative multi-omics data elucidating the biosynthesis and regulatory mechanisms of furanocoumarins in *Angelica dahurica*”. We gratefully thank the editor and the three reviewers for their time spent on these constructive comments. We have conducted an extensive revision by addressing their comments, especially regarding the evolution of the furanocoumarin synthesis pathway. Below the comments of the reviewers are our point-by-point responses. The comments from reviewers are in blue below, and our responses are in black fonts. We have also included a tracked version of our manuscript with edits to address reviewers’ concerns in red.

Thank you kindly,

Dr. Li Wang

Reviewer #1

Comments to the Author:

Thank you for submitting your manuscript entitled "Integrative multi-omics data elucidating the biosynthesis and regulatory mechanisms of furanocoumarins in Angelica dahurica" to Communications biology. The paper mainly constructed a chromosome-level genome for A. dahurica and quantified the concentration dynamics of 17 coumarins across six developmental stages of the root. The identification of specific CYP450 enzymes involved in bergaptol and xanthoxol biosynthesis, along with the evolutionary history of these enzymes, offers valuable information for enhancing the insights into furocoumarins biosynthetic pathway and the epigenetic regulation of metabolite biosynthesis. The manuscript was rich and reliable in content. I suggest this manuscript can be accepted after the following minor revisions. More exhaustive comments are as follows.

Response: We sincerely appreciate your careful and thoughtful review of our manuscript! We have extensively revised the manuscript according to these suggestions. We believe it will largely improve the manuscript.

Comment 1: *In lines 138-140. "For genome annotation, transcriptomes of 11 tissues, including roots from seven developmental stages, stems, leaves, and flowers were sequenced. By combining ab initio, homology-based and transcriptome-based approaches". This part should be moved to materials and methods more appropriately.*

Response: Thank you for your feedback. I agree that moving the sentence to the Materials and Methods section (line 514-518, page 18) would be more appropriate. It now reads,

"To predict coding gene structures within the repeat-masked genome, we utilized a multifaceted approach that included ab initio predictions, evidence from homologous proteins, and transcriptome data of 11 tissues, including roots from seven developmental stages, stems, leaves, and flowers were sequenced (Table S2)."

Comment 2: *In line 154. "D. carota, L. chuanxion and A. sinensie." The Latin name of the species should be fully listed with its genus when it first appears in the text.*

Response: Thank you for your careful attention. We have made the necessary corrections to ensure that all latin names are presented accurately and consistently throughout the text.

Comment 3: *In lines 161-162. "The number of expanded gene families in A. dahurica was almost twice as that in D. carota, A. sinensis and L. chuanxiong." How do the authors interpret the implications of having nearly twice as many expanded gene*

families in A. dahurica compared to other species? What are the possible reasons for the difference? Discussion on potential adaptive advantages of these expansions would enhance the understanding. This is just a suggestion.

Response: Thank you for your advice! As the expanded gene families were likely originated from gene duplication. Thus, we classified the duplicated genes within expanded gene families of *A. dahurica* into five categories: whole-genome duplication (WGD), tandem duplication (TD), proximal duplication (PD), transposed duplication (TRD), and dispersed duplication (DSD). Among these, PD was the predominant type (27.89%, 1807 genes), followed by WGD (1726 genes, 26.64%) and TD (1190 genes, 18.37%). PD, TD, and DSD genes were specifically enriched in secondary metabolic pathways, including phenylpropanoid biosynthesis (the shared upstream pathway of coumarin biosynthesis), flavonoid biosynthesis, and benzoxazinoid biosynthesis. These secondary metabolites enhance environmental adaptability, suggesting that expanded gene families in *A. dahurica* plays a critical role in its adaptation to variable environments.

Fig S3 Classification of duplicated genes within expanded gene families in *A. dahurica*. WGD, whole-genome duplication, TD, tandem duplication, PD, proximal duplication, TRD, transposed duplication, DSD, dispersed duplication.

Fig. S4 Kyoto Encyclopedia of Genes and Genomes (KEGG) enrichment analyses of different types of duplicated genes within expanded gene families. The enriched terms with adjusted $P < 0.05$ are presented. Color of the bubbles indicates statistical significance of the enriched terms; size of the bubbles indicates number of genes within the term.

We have added relevant descriptions in the discussion section (line 332-336, page 12). It now reads,

“Gene family expansion in *A. dahurica* was mainly driven by PD (27.89%), WGD (26.64%), and TD (18.37%); while PD and TD were specifically and significantly enriched in secondary metabolic pathways, including phenylpropanoid biosynthesis (the shared upstream pathway of coumarin biosynthesis) and flavonoid biosynthesis, which were crucial for environmental adaptation.”

*Comment 4: In line 194. The findings indicate a decline in overall FC concentration despite increased root biomass. What are the potential biological implications of this observation? How are these substrate concentration changes related to FCs accumulation? Discussing this could provide a deeper insight into the metabolic strategy of *A. dahurica* during root development.*

Response: Thank you for your advice. We apologize for not clearly describing the potential biological implications. On the one hand, FCs are a class of defensive metabolites primarily distributed in the phloem of roots rather than the xylem, enabling timely and effective responses to various pathogens and phytophagous insects. However, as root enlargement leads to an increase in the proportion of xylem, the proportion of phloem becomes relatively contracted, resulting in a decrease in coumarin

concentration per unit volume. On the other hand, the biosynthesis of secondary metabolites, including FCs, requires significant energy expenditure. As juvenile plants grow stronger and their tolerance to pathogens and phytophagous insects increases, they tend to balance energy allocation between growth and defense, potentially regulating FC biosynthesis. It might lead to an overall decline in FC concentration. Given that the diameter and biomass of *A. dahurica* roots continue to increase during the S1-S5 period, the total coumarin content may still rise. Considering the defensive role of FCs, applying moderate stress during *A. dahurica* growth could enhance coumarin accumulation in the roots, though careful attention must be paid to its potential impact on yield. And we have included this discussion in the revised manuscript (line 358-376, page 13). It now reads,

“On the one hand, FCs are a class of defensive metabolites primarily distributed in the phloem of roots rather than the xylem, to enable timely and effective responses to various pathogens and phytophagous insects⁴⁷. Thus, the observed decline may be attributed to root enlargement, which increases the proportion of xylem and causes the proportion of phloem to become relatively contracted, leading to a reduction in coumarin concentration per unit volume. On the other hand, biosynthesis of secondary metabolites demands significant energy expenditure and plants tend to balance energy consumption between growth and defense^{1,248,49}. As juvenile plants mature and their tolerance to pathogens and phytophagous insects increases, the observed decrease in FCs concentration in the roots compared to the seedling stage may reflect a regulatory strategy employed by the plant to optimize resource allocation between growth and defense. However, given that the root diameter of *A. dahurica* continued to increase during the S1-S5 period, with a corresponding increase in root biomass, harvesting during the S5 period remains sensible. The sustained high levels of imperatorin and isoimperatorin observed throughout root development may reflect their pivotal role in biodefense. Given the defensive function of FCs, the application of moderate stress during the developmental stages of *A. dahurica* may promote coumarin accumulation in the roots; however, its potential effects on yield should be carefully evaluated.”

Comment 5: In line 200. The manuscript mentions the integration of multi-omics data but lacks details about how different omics layers were combined. It is suggested to clarify their approach to integrating transcriptomic, metabolomic, and any other omics data in the methods section. The correlation analysis used to connect gene expression levels and metabolite content is significant, but the statistical software used to assess this correlation was not specified in the methods section.

Response: Thank you for your comments! To identify potential CYP450 genes in *A. dahurica*, we employed a three-step approach to systematically narrow down the candidate gene set. First, we selected genes with conserved domain (Pfam ID: PF00067), sequence similarity over 40% compared with CYP71AZ4 from *Pastinaca sativa* as a query, and filtered out genes with protein sequences < 300 amino acids or gene expression level (FPKM) = 0. It yielded a putative CYP450 gene set comprising 310 genes. This step integrated genomic sequences with transcriptomic expression to refine the candidate gene set. Second, a phylogenetic analysis of the 310 genes,

CYP450 protein sequences in *Arabidopsis thaliana*, as well as sequences verified to be involved in FC biosynthesis. 25 genes within the CYP71 clade clustered with CYP71AZ4 were identified as preliminary candidates for P8H and P5H. Finally, by integrating transcriptomic and metabolomic data, we calculated the Pearson correlation between the expression levels of these genes and the corresponding metabolite contents across various tissues and root developmental stages. This analysis obtained two genes strongly correlated with xanthotoxol content (Pearson coefficient > 0.75) and four genes correlated with bergaptol content, which were subsequently prioritized for experimental validation. We further emphasize Pearson correlation in our method (line 581-586, page 20). It now reads,

“We further calculated the Pearson correlation coefficients between gene expression levels and secondary metabolite contents from multiple tissues and root developmental stages of the 25 preliminary candidates in *A. dahurica*. Ultimately, we obtained two genes strongly correlated with xanthotoxol content ($r > 0.75$) and four genes correlated with bergaptol content, which were subsequently prioritized for experimental validation.”

Comment 6: In line 212. All gene names should be italic. Check and modify the rest of the manuscript.

Response: Thank you for noting the formatting issue. We have reviewed the entire manuscript and ensured that all gene names are italicized. At the same time, when describing the protein sequence, we still keep the name of the protein sequence non-italicized, for example (line 225-227, page 9):

“we collected three groups of CYP71 protein sequence: 1) three verified enzyme from *A. dahurica* (CYP83F95, CYP71AZ18 and CYP71AZ19);”

Comment 7: In lines 216-217. The authors report that AdP8H1, AdP8H2, and AdP5H1 exhibited catalytic activities in a tobacco transient expression system, but the function of these genes in this pathway is not specifically elucidated here. It is suggested that the specific roles of these enzymes in the metabolism of xanthotoxol and bergaptol should be elaborated.

Response: Thank you for your constructive feedback. We have revised the text to specify their ability to promote conversions in a tobacco transient expression system (line 215-220, page 8). It now reads,

“In the tobacco transient expression system, *AD04G02371*, and *AD043G00078* were found to catalyzes hydroxylation at C-8 positions of psoralen into xanthotoxol, while *AD04G02366* react on hydroxylation at C-5 positions, significantly enhancing the conversion of psoralen into bergaptol (Fig. 3D). These genes were subsequently named *CYP71AZ19*, *CYP83F95*, *CYP71AZ18*, respectively.”

Comment 8: In lines 219, 222, and 225. “*A.dahurica-specific*” should be changed to “*A. dahurica-specific*”. “*A.dahurica genes*” should be changed to “*A. dahurica genes*”. “(3) *CYP71As and CYP71Bs*” should be changed to “(3) *CYP71As and CYP71Bs*”. Check and modify the rest of the manuscript.

Response: Thank you for your careful review. We have addressed the formatting issues you raised and corrected all similar instances in the manuscript.

Comment 9: In line 232. This notes that both *CYP71AZ* and *CYP83F* subfamilies underwent lineage-specific duplication events in *Apiaceae*. How might they contribute to the functional diversity of these gene families within the *Apiaceae*? The analysis states that there is no observed collinearity for *CYP71AZ19* with the other genes in *Apiaceae* species. What are the possible reasons for this absence of collinearity?

Response: Thank for your comment! We constructed the phylogenetic relationships of *CYP71AZ* and *CYP83F* from multiple species in the *Apiaceae*, *Rutaceae*, *Moraceae*, and *Fabaceae* families. It demonstrated that no genes from *Rutaceae*, *Moraceae*, or *Fabaceae* clustered within the same clades as *CYP71AZ* and *CYP83F* from *Apiaceae*, suggesting that these two CYP450 families underwent lineage-specific duplication events unique to *Apiaceae* post divergence from *Rutaceae*, *Moraceae*, and *Fabaceae*. Gene duplication is a key driver of functional diversity by providing additional gene copies that serve as foundation for the evolution of novel functions. Through subfunctionalization or neofunctionalization, the duplicated gene copies can acquire new roles, thereby facilitating the biosynthesis of FC.

Through the analysis of expanded gene families in *A. dahurica*, we identified *CYP71AZ18* and *CYP71AZ19* as products of a proximal duplication event. Notably, *CYP71AZ18* showed collinearity with *AS02G00065* in *A. sinensis*, *Pp8G3485* in *Peucedanum praeruptorum*, and *Ps1G0267* in *Pastinaca sativa*; while collinearity with *CYP71AZ19* was absent in other *Apiaceae* species. These results suggest that *CYP71AZ19* likely arose from a proximal duplication of *CYP71AZ18* concomitant with the speciation of *A. dahurica*. We have clarified it in the results (line 240-250, page 9). It reads,

“Based on the synteny analysis, *CYP71AZ18* was collinear with *AS02G00065* in *A. sinensis*, *Pp8G3485* in *P. praeruptorum*, *Ps1G0267* in *P. sativa*, respectively (Fig. 5B). However, collinearity with *CYP71AZ19* was absent in the above-mentioned *Apiaceae* species (Fig. 5B). Coupled with the finding that *CYP71AZ18* and *CYP71AZ19* are products of proximal duplication (Fig. S3), we inferred that *CYP71AZ19* originated from a proximal duplication event of *CYP71AZ18* concomitant with the speciation of *A. dahurica*.”

Comment 10: In line 289. The conclusions highlight the importance of ACRs in regulating gene expressions, particularly in phenylpropanoid biosynthesis. It is suggested that a brief overview of the roles of key genes within these pathways to

enhance the reader's understanding of the biological implications?

Response: Thank you for your advice! We added a description of the relationship between the phenylpropanoid and FC biosynthesis pathway (line 294-299, page 11). It now reads,

“The biosynthesis of FC begins with phenylalanine (Fig. 1), which undergoes a series of deamination, hydroxylation, and cyclization reactions to produce umbelliferone, the shared substrate in the FC biosynthetic pathway. Taken together, ACRs play a critical role in regulating gene expression, particularly within the phenylpropanoid biosynthesis pathway in roots, ultimately regulating the accumulation of FCs.”

Comment 11: In line 309. The findings indicate tissue-specific expression patterns of genes involved in FC biosynthesis in A. dahurica. Differences between gene expression patterns and metabolite content were noted. What is the possible reason for this observation?

Response: Thank you for pointing this out. The concentration of FCs is influenced by their biosynthesis, degradation, and transportation. Expression levels of related genes reflect FC biosynthesis, but may not precisely match metabolite levels. This underscores the importance of integrating multi-omics data analysis when screening candidate genes in the FC biosynthesis pathway, ensuring that functional genes are not inadvertently excluded by relying solely on a single threshold. We added relevant content into the discussion (line 429-433, page 16). It now reads,

“The concentration of FCs is influenced by their biosynthesis, degradation, and transportation, and thus may not fully correlate with the expression levels of genes catalyzing these processes. ATAC-seq data can serve as a complementary tool for screening candidate genes, to enhance screening efficiency and prevent the exclusion of functional genes due to overly stringent criteria.”

Comment 12: In lines 494, 503. “Daucus carota ” and “B. chinense and D. carota ”. The species names should be in italics. Check and modify the rest of the manuscript.

Response: We appreciate your attention to such details. We have addressed the formatting issue you raised, and ensured that all latin names in the manuscript are correctly formatted in italics.

Comment 13: In lines 505, 508. CAFE5 and MCscanX. Please add the software version information.

Response: Thank you for your feedback. We have added the software version for CAFE5 v5.0 and MCscanX v1.1.11 in lines 537 and 540.

Comment 14: In line 515. “Kyoto Encyclopedia of Genes and Genomes (KEGG) ”.

As already mentioned, it can be abbreviated.

Response: We appreciate your attention to such details. We have addressed the issues you raised in line 515 and ensured consistency in abbreviation usages throughout the manuscript.

Comment 15: *In line 597. “and Angelica sinensis ” . “and ” should be in body, not italics. Check and modify the rest of the manuscript.*

Response: Thank you for bringing this issue to our attention. We have corrected the formatting issue you raised and also thoroughly reviewed the manuscript to ensure the proper use of italic formatting.

Comment 16: *There are many mistakes in the references. Please ensure the accuracy and completeness of the references. For example:*

In lines 657, 728. “Nat Methods ” and “Plant J ” . Please check the format of the journal, not abbreviations. Check and modify the rest of the manuscript.

In line 694. “Aedes aegypti” should be changed to “Aedes aegypti ” . “Coptis chinensis ” in Line 739, and “k-mers ” in Line 805, etc. Please check the article title.

In line 694. “Schulz, A.J.P.o.t.N.A.o.S. ” The author information is incorrect. The literature lacks a journal title. Please check and modify it.

Some punctuation mistakes should be checked carefully, such as "The family of furocoumarins:." should be "The family of furocoumarins:" in Line 757, etc.

Response: Thank you for pointing out the mistakes in the references. We have thoroughly checked and modified the entire reference list to ensure that all references are correct and complete. We have also double-checked the citation format to make sure it is consistent throughout the manuscript.

Comment 17: *In line 991. “Ethics declarations ” . There is a lack of content here.*

Response: Thank you for bringing this to our attention. We would like to clarify that the 'Ethics declarations' section of our manuscript does indeed contain the 'Competing interests' statement 'The authors declare no competing interests', which is in compliance with the practices of this journal.

We apologize if the formatting of our manuscript caused any confusion. We have reviewed the section to ensure that it is correctly formatted and easily identifiable. Thank you for your time and effort in reviewing our work, and we appreciate your understanding.

Comment 18: *In line 999. “Supplemental information titles and legends ” . Many*

species name format errors, please check carefully.

Response: We appreciate your time and effort in helping us to improve the accuracy of our manuscript. We have carefully reviewed the "Supplemental information titles and legends" section and corrected any latin name errors. We have also double-checked the rest of the manuscript to ensure that all latin names are formatted correctly.

Comment 19: *The “5’ UTR” positions overlap in Fig. 5C, please adjust it.*

Response: Thank you for pointing this out. We have adjusted it to ensure that the positions are clearly distinguishable and do not overlap in Fig. 5C (current Fig. 6C).

Reviewer #2

Comments to the Author:

This paper present results on the biosynthesis of furanocoumarins in Apiaceae plants. These molecules constitute an interesting subject of evolutionary escalation between plants and insects as has been long ago shown by the groups of May Berenbaum and Mary Schuler. The data were obtained from Angelica dahurica, an important plant in traditional Chinese medicine. The main results are (1) the identification of CYP71AZ18, a gene encoding psoralen 5-hydroxylase, a missing step in the synthesis of 5-hydroxy psoralen derivatives,(2) the evolutionary origin of CYP71AZ19, a P450 carrying psoralen 8-hydroxylase activity, from a duplication/neofunctionalization event of CYP71AZ18, (3) the correlation of chromatin accessibility with the expression of genes involved in the synthesis of furanocoumarins and the concentrations of these compounds found in different plant tissues.

Considering that many P450s have now been described in the furanocoumarin pathway and that neofunctionalized duplication has already been demonstrated in furanocoumarins produced by Apiaceae plants, the main novelty of this manuscript lies in (3). The authors have produced a large quantity of data, from genome sequencing of the plant to transcriptional analyses, from characterisation of novel orthologs and paralogs of the CYP71AZ subfamily to furanocoumarin concentrations assessed at different periods of the growth cycle. My main concern is that the manuscript just scratches the surface of different topics that could have been more studied in more depth with the same dataset. I personally expected a more detailed study of one of the 3 aforementioned topics. Remarks for improving the manuscript as it is:

Response: Thank you very much for your valuable suggestions and thoughtful comments on our manuscript! Your feedback has been incredibly helpful, and we truly appreciate the time and effort you've taken to provide such detailed insights. We focused more on the evolution of the FC biosynthetic pathway. We have also added new analyses, comparing genes involved in FC biosynthesis both within Apiaceae and between Apiaceae and three other plant families known for high FCs levels. The

manuscript has been extensively revised in accordance with these suggestions, and we believe these changes significantly enhance the quality and impact of the work.

Comment 1: L43: the term "charm" is certainly inappropriate here.

Response: Thank you for your valuable feedback. In the revised manuscript, we have removed this statement.

Comment 2: L47-52: description of the interest of furanocoumarins in medicine and dermatology is outdated. Since the early 90s, these compounds have been banned from cosmetic products because they are highly carcinogenic, and directly correlated with the rate of malignant melanoma in humans. Psoriasis is now treated with new small molecules such as Deucravacitinhib or anti-IL17/23 antibodies for the most severe cases.

Response: Thank you for your reminder regarding the medicinal value of FCs. We sincerely apologize for not incorporating the latest literatures. In the revised version, we have redirected our focus to the evolutionary and ecological significance of FCs rather than their pharmacological activities. Consequently, we have removed detailed descriptions of FCs' therapeutic applications in diseases from the Introduction section (line 46-55, page 3). It now reads,

“Plants have evolved the ability to produce specialized metabolites as an adaptive response to environment. Among these metabolites, furocoumarins (FCs) are defense chemicals against various bioaggressors. FCs play a crucial ecological role by defending against pathogens, inhibiting germination of competing plants, and deterring herbivores¹. Additionally, FCs exhibit substantial pharmacological efficacy, encompassing anticancer, antimicrobial, and anti-inflammatory effects in human^{2,3}. FCs were exclusively identified in a few phylogenetically distant plant families, including the Apiaceae, Rutaceae, Fabaceae and Moraceae. The sporadic FCs distribution across angiosperm phylogeny implies multiple independent origins of FC biosynthesis in the four families^{4,5}.”

Comment 3: L56: the two proposed references (72 and 74) relating to the biosynthetic genes described to date are confusing. Indeed, none of them correspond to articles describing new biosynthetic enzymes. 72 concerns the metabolic engineering of the furanocoumarin pathway and 74 is a book chapter referring to the diversity of furanocoumarins found in the plant kingdom. Instead, cite here original work on the discovery of biosynthetic enzymes.

Response: Thank you for pointing this out. We have added references to the original work on the discovery of biosynthetic enzymes. The references are as follows:

5 Munakata, R. *et al.* Parallel evolution of UbiA superfamily proteins into aromatic O-prenyltransferases in plants. *Proc Natl Acad Sci U S A* **118**, doi:10.1073/pnas.2022294118 (2021).

- 8 Jian, X. *et al.* Two CYP71AJ enzymes function as psoralen synthase and angelicin synthase in the biosynthesis of furanocoumarins in *Peucedanum praeruptorum* Dunn. *Plant Mol Biol* **104**, 327-337, doi:10.1007/s11103-020-01045-4 (2020).
- 9 Zhang, Y., Bai, P., Zhuang, Y. & Liu, T. Two O-Methyltransferases Mediate Multiple Methylation Steps in the Biosynthesis of Coumarins in *Cnidium monnieri*. *J Nat Prod* **85**, 2116-2121, doi:10.1021/acs.jnatprod.2c00410 (2022).

Additionally, in describing the CYP450 genes involved in FC biosynthesis, we have included references to all experimentally validated CYP450 enzymes in the FC biosynthetic pathway to date in line 65-67. It now reads,

To date, CYP450s involved in FC biosynthesis belong to the CYP71 clan and are responsible for furan-ring formation and hydroxylation, leading to the diversification of FCs (Fig. 1)^{1,8,13-20}.

Reference:

- 1 Villard, C. *et al.* A new P450 involved in the furanocoumarin pathway underlies a recent case of convergent evolution. *New Phytol* **231**, 1923-1939, doi:10.1111/nph.17458 (2021).
- 8 Jian, X. *et al.* Two CYP71AJ enzymes function as psoralen synthase and angelicin synthase in the biosynthesis of furanocoumarins in *Peucedanum praeruptorum* Dunn. *Plant Mol Biol* **104**, 327-337, doi:10.1007/s11103-020-01045-4 (2020).
- 13 Larbat, R. *et al.* Isolation and functional characterization of CYP71AJ4 encoding for the first P450 monooxygenase of angular furanocoumarin biosynthesis. *J Biol Chem* **284**, 4776-4785, doi:10.1074/jbc.M807351200 (2009).
- 14 Krieger, C. *et al.* The CYP71AZ P450 Subfamily: A Driving Factor for the Diversification of Coumarin Biosynthesis in Apiaceous Plants. *Front Plant Sci* **9**, 820, doi:10.3389/fpls.2018.00820 (2018).
- 15 Huang, X.-C. *et al.* The gradual establishment of complex coumarin biosynthetic pathway in Apiaceae. *Nature Communications* **15**, doi:10.1038/s41467-024-51285-x (2024).
- 16 Larbat, R. *et al.* Molecular cloning and functional characterization of psoralen synthase, the first committed monooxygenase of furanocoumarin biosynthesis. *J Biol Chem* **282**, 542-554, doi:10.1074/jbc.M604762200 (2007).
- 17 Kruse, T. *et al.* In planta biocatalysis screen of P450s identifies 8-methoxypsoralen as a substrate for the CYP82C subfamily, yielding original chemical structures. *Chem Biol* **15**, 149-156, doi:10.1016/j.chembiol.2008.01.008 (2008).
- 18 Limones-Mendez, M. *et al.* Convergent evolution leading to the appearance of furanocoumarins in citrus plants. *Plant Sci* **292**, 110392, doi:10.1016/j.plantsci.2019.110392 (2020).
- 19 Wang, K. *et al.* Three types of enzymes complete the furanocoumarins core skeleton biosynthesis in *Angelica sinensis*. *Phytochemistry* **222**, 114102, doi:10.1016/j.phytochem.2024.114102 (2024).
- 20 Zhao, Y. *et al.* Two types of coumarins-specific enzymes complete the last missing steps in pyran- and furanocoumarins biosynthesis. *Acta Pharm Sin B* **14**, 869-880, doi:10.1016/j.apsb.2023.10.016 (2024).

Fig. 1 Schematic representation of the FC biosynthetic pathway and the validated catalytic enzymes in the Apiaceae, Moraceae, Rutaceae, and Brassicaceae families. U6P: umbelliferone-6-prenyltransferase; U8P: umbelliferone-8-prenyltransferase; MS: marmesin synthetase; CS: columbianetin synthetase; PS: psoralein synthetase; AS: angelicin synthetase; P5H: psoralein-5-hydroxylase; P8H: psoralein-8-hydroxylase; PT: prenyltransferase; BMT: bergaptol O-methyltransferase; XMT: xanthotoxol O-methyltransferase; B8H: bergapten-8-hydroxylase; X5H: xanthotoxin-5-hydroxylase; OMT: O-methyltransferase.

Comment 4: L62-71: Rather than citing 4 references at the end (36, 41, 44, 84), it is better to insert them where they are relevant in this paragraph.

Response: Thank you for your advice! In our response to comment 3, we summarized and cited all references included in Fig. 1. Additionally, when providing examples, we have inserted relevant references in appropriate locations (line 69-73, page 3). It now reads,

“For instance, the conversion of xanthotoxin to 5-hydroxyxanthotoxin is catalyzed by CYP71AZ¹⁴, CYP82D¹⁸ and CYP71B²¹ families in Apiaceae, Rutaceae, and Moraceae, respectively. Similarly, the marmesin synthetase identified in Apiaceae and Moraceae were from CYP450 families: CYP736A¹⁹ and CYP76F¹.”

Comment 5: L106: spelling error for imperatorin and isoimperatorin.

Response: Thank you for pointing this out. We have corrected it accordingly (line 95, page 4).

Comment 6: L130: avoid judging the anchoring rate. It's not impressive, it's just the way it is.

Response: Thank you for the comment. We have modified the relevant text regarding the anchoring rate to exclude any sense of judgements and maintain neutrality.

Comment 7: L138 I'm not convinced by the term "developmental stage" which should imply a different physiological stage. Here it's more about "sampling dates" not necessarily related to ontogeny. Also check the number of sampling dates because I count 6 and not 7 in Fig. 2.

Response: Thank you for your feedback. The six sampling dates shown in Fig. 2 represent the monthly root sampling events from May 2021 to October 2021 for *A. dahurica* seedlings planted in March 2021. Please note that these six dates do not include the additional sampling of five distinct tissues (including roots) from a different set of samples planted in August 2020. We have modified "developmental stage" in line 179. It now reads,

"To investigate the accumulation pattern of FCs in the root at different sampling dates of *A. dahurica*"

Upon review, the sample size for S1 was seven, with root diameters of 1.05, 1.2, 1.35, 1.4, 1.4, 1.5 and 1.9. Due to the identical root diameter (1.4) of two samples, their data points overlapped entirely in the scatter plot, leading to the appearance of only six distinct samples in Fig. 2B (now Fig. 3B).

*Comment 8: L154: please write the full genus name of each plant species. I didn't know what *I. chuanxiong* was before going to Fig. 1.*

Response: Thank you for your suggestion. We have made necessary revisions to ensure that the full genus name of each plant species appears before any abbreviations are used.

Comment 9: L195: the decrease in furanocoumarin concentration in late harvested roots could be due to a dilutive effect related to the increase in biomass while furanocoumarin synthesis could have stopped at this stage. Just my opinion.

Response: We appreciate your insightful speculation. The observed decrease in coumarin concentration might result from a combination of a dilutive effect associated with increased biomass and a potential decline in coumarin biosynthesis capacity. Given that the three functionally-verified genes catalyzing FC biosynthesis are expressed throughout the S1-S5 stages, it is unlikely that biosynthesis has ceased

completely. By addressing the comment four of the Reviewer one, we have expanded relevant discussions (line 358-369, page 13). It now reads,

“On the one hand, FCs are a class of defensive metabolites primarily distributed in the phloem of roots rather than the xylem, to enable timely and effective responses to various pathogens and phytophagous insects⁴⁷. Thus, the observed decline may be attributed to root enlargement, which increases the proportion of xylem and causes the proportion of phloem to become relatively contracted, leading to a reduction in coumarin concentration per unit volume. On the other hand, biosynthesis of secondary metabolites demands significant energy expenditure and plants tend to balance energy consumption between growth and defense^{48,49}. As juvenile plants mature and their tolerance to pathogens and phytophagous insects increases, the observed decrease in FCs concentration in the roots compared to the seedling stage may reflect a regulatory strategy employed by the plant to optimize resource allocation between growth and defense.”

Comment 10: L212-217: this paragraph is just too straight to the point. First, the selected genes are only candidates, not P8H or P5H as stated here. Also, please use the international codification for genes (script letters) and protein sequences (capital letters). The real status of these sequences should be clearly written here (orf? cDNA? etc.

Response: We sincerely appreciate your careful and thoughtful review of our manuscript! We removed the naming of *AdP8H1* and replaced it with the gene id in line 211-215. It now reads,

“Ultimately, based on the correlation between FPKM and metabolite levels in diverse tissues ($r > 0.75$, Table S7), we selected two P8H candidates (*AD04G02371* and *AD03G00078*), and four P5H candidates (*AD04G02366*, *AD08G00823*, *AD03G00273* and *AD04G04017*) for downstream experimental validation (Fig. 4B).”

Comment 11: Part on chromatin accessibility (L291-303). This part is quite confusing because Fig. 6 presents a lot of data that is not commented by the authors. Why comment only on Fig. 6A? After reading this part, it is not clear whether chromatin accessibility is positively correlated or not with transcriptional data of biosynthetic genes. Fig. 6C contains many cDNA sequences presented under each enzymatic step but it is not clear whether these activities have been experimentally assigned or are just hypotheses based on sequence similarities.

Response: Apologies for the lack of clarity in the layout of this section. The paragraph below lines 291-303 describes Figure 6C (former lines 305-314, current lines 355-368). Among the three genes validated in this study, *CYP71AZ18/19* exhibited consistent tissue specificity in terms of chromatin accessibility and gene expression levels. However, *CYP83F95* did not, which may be attributed to its low expression levels. Figure 6C (current Figure 7C) presents candidate genes identified based on sequence similarity. As genes with ACRs generally show higher expression, chromatin

accessibility may help us further refine the list of candidate genes. We have included this interpretation in the results section (line 315-328, page 12). It now reads,

“In order to investigate how chromosome accessibility might impact expression of biosynthetic genes in the complete FC biosynthetic pathway, we treated genes exhibiting over 75% homology to queries (genes with experimentally validated functions; Table S8) as candidate genes involved in FC biosynthetic pathway, and quantified their expression levels and chromatin accessibility. It revealed that while the expression patterns of these genes in roots and leaves did not consistently mirror the profile of metabolite content, they were consistently correlated with chromatin accessibility (Fig. 6C). Notably, as long as the identified candidate genes exhibited chromatin accessibility, expression levels and downstream product concentrations would be observed. These genes represented promising candidates for subsequent experimental validation. For instance, among the five candidate genes identified for the conversion of xanthotoxol to imperatorin, *AD01G04637* emerged as a top priority for experimental validation due to its consistent profile in chromatin accessibility, gene expression, and metabolite content distribution.”

Comment 12: L43: L299-303: There is probably much more to discover in the genomic data regarding transcription factors than just the abscisic acid and WRKY factors mentioned here. This question of transcriptional regulation of the furanocoumarin pathway is completely unresolved to date.

Response: Thank you for your constructive suggestions. We agree with you that the regulation of the FC biosynthesis pathway is much complex, and the identified ABI3 and WRKY transcription factors just represent the tip of an iceberg. We have included additional motifs in the Fig S9 as a candidate dataset for further investigation into the regulation of the FC biosynthesis pathway.

Fig. S9 Motifs potentially regulating FC biosynthesis, enriched in DARs that are upregulated in the roots of *Angelica dahurica* relative to the leaves.

Comment 13: General comment on the figures: some histograms are too small. The name of the furanocoumarin standards should be added in Fig.3D.

Response: Thanks for raising this question. We have modified accordingly.

Reviewer #3

Comments to the Author:

Integrative multi-omics data elucidating the biosynthesis and regulatory 2 mechanisms of furanocoumarins in Angelica dahurica" by Ji et al., reported the assembly of a chromosome-level genome for a famous perennial Chinese traditional medicinal plant Angelica dahurica and explored the dynamics of 17 coumarins across six developmental stages of roots, which are usually used in clinical treatments, as they are enriched with bioactive volatile terpenoids and coumarins.

The authors in a genome research group have published several genome assemblies of same plant family. All of them contain high levels of volatile terpenoids and coumarins, that might be responsible for their pharmacological activity and supporting their clinical applications. In this A.dahurica genome, the authors particularly focused on

furocoumarins (FCs) biosynthesis pathway and structural genes. They proposed CYP71AZ18 was involved in the biosynthesis of bergaptol, whereas CYP71AZ19 and CYP83F95 contributed to the biosynthesis of xanthotoxol, with several pieces of biochemical evidence. and also analyzed the evolutionary relationships among these P450 genes, in Angelica genus, etc.

While the manuscript about the genome sequence analysis and assembly quality are routinely well done, also proposed genes from researching the genome and transcriptomic databases in the constructed biosynthesis pathways, I see some weak points:

Response: We sincerely appreciate your kind review and insightful comments. We have revised our manuscript based on your suggestions. We have added the evolutionary analysis of CYP71AZ and CYP83F both within the Apiaceae and among the four angiosperm families (i.e., Moraceae, Rutaceae, Apiaceae and Fabaceae). Please see our detailed response below.

***Comment 1:** The authors may want to check carefully on the characterization of CYP71AZ19/18 and CYP83F95, for reactions and LC-MS examination of the end products in the reactions.*

Response: We sincerely appreciate your careful and thoughtful review of our manuscript! We have carefully reviewed the characterization of CYP71AZ19/18 and CYP83F95, including their catalytic reactions and LC-MS analysis of the end products. The experimental data confirm that CYP71AZ19 and CYP71AZ18 catalyze the hydroxylation of psoralen at the C-8 position to produce xanthotoxol, while CYP83F95 facilitates hydroxylation at the C-5 position, leading to bergaptol. The LC-MS of the reaction products have been re-examined. We believe these results robustly support our conclusions.

***Comment 2:** By comparison of these genes in other Angelica species that have been sequenced, how they are different from evolutionary genomic perspective? The enriched FC in A. dahurica could be reflected from the gene expansion of duplication of key genes, as compared with A. sinensis, D. carota, L. chuanxiong in Apiaceae family or from four plant families: Apiaceae, Rutaceae, Moraceae, and Fabaceae that contain higher level of FC, their indeed divergent or convergent mechanics are? from the perspective of the P450 gene evolution.*

Response: We sincerely appreciate your careful and valuable comments! To explore the evolutionary history of FC biosynthesis in a large phylogenetic framework, we analyzed CYP71 protein sequences from three groups: 1) three verified enzymes from *A. dahurica* (CYP83F95, CYP71AZ18, CYP71AZ19); 2) outgroup representatives from *A. thaliana*; and 3) CYP71 homologs from five Apiaceae species, one Rutaceae, one Moraceae, and one Fabaceae species, identified via Blastn with >55%

sequence identity. Phylogenetic analysis divided CYP71s into three clades: Clade I included genes from Rutaceae, Moraceae, and Fabaceae; Clade II comprised the CYP71AZ subfamily in Apiaceae; and Clade III contained the CYP83F subfamily, including CYP83F95. The distinct clustering of Apiaceae genes suggests lineage-specific duplication and diversification of CYP71AZ and CYP83F subfamilies, enabling their catalytic roles in FC biosynthesis. In contrast, although P5H and P8H have been identified in *A. dahurica*, no orthologous genes have been found in the other three families, indicating that these steps are likely catalyzed by entirely different CYP450 enzyme families in those lineages.

Within the Apiaceae family, the number of CYP71AZ and CYP83F gene copies (with length of amino acid sequences > 100) varies among species: 27 in *A. dahurica*, 18 in *A. sinensis*, 29 in *L. chuanxiong*, 20 in *P. praeruptorum*, 19 in *P. sativa*, and 11 in *D. carota*, in which no FC were reported. It is possible that CYP71AZ and CYP83F genes were duplicated post the divergence between the five species with high FC content and *D. carota*, as *D. carota* represented a more basal lineage compared to the other five species in Apiaceae. To sum it up, the identification and evolutionary history of CYP71AZ and CYP83F provides a novel example of how CYP450 family expands and evolves, leading to rich content and abundant diversity of FCs. The results and discussion of the evolutionary analyses of CYPs are added in lines 224-243, lines 378-401, respectively. It now reads,

“To investigate the evolutionary history of FC biosynthesis pathway in a large phylogenetic framework encompassing the four high-FC angiosperm families, we collected three groups of CYP71 protein sequences: 1) three verified enzymes from *A. dahurica* (CYP83F95, CYP71AZ18 and CYP71AZ19); 2) outgroup representatives, CYP51G1 and CYP71A12 from *A. thaliana*; 3) CYP71 homologs from five Apiaceae species (*A. sinensis*, *L. chuanxiong*, *Peucedanum praeruptorum*, *Pastinaca sativa* and *D. carota*), one Rutaceae species (*Citrus limon*), one Maraceae species (*Morus notabilis*) and one Fabaceae species (*Medicago truncatula*). The phylogenetic tree of CYP71s was divided into three clades (Fig. 5A). Notably, genes from Rutaceae, Moraceae, and Fabaceae formed a basal clade (Clade I), distinct from the CYP71AZ and CYP83F subfamilies in Apiaceae. Clade II comprised the CYP71AZ subfamily in the Apiaceae, including CYP71AZ18 and CYP71AZ19; while Clade III encompassed the CYP83F subfamily in the Apiaceae, including CYP83F95 (Fig. 5A). Interestingly, *M. notabilis* (without FCs) had three homologs, whereas *Ficus carica* (with FCs) lacked any homolog. Together with the observation that homologous genes of the other three families were clustered into a distinct clade from those in Apiaceae, we proposed that the function of CYP71AZ and CYP83F subfamilies was not conserved among high-FC families, and they underwent lineage-specific evolution to acquire the catalytic function in Apiaceae. In contrast, the corresponding step might be catalyzed by enzymes from distinct subfamilies in the other three families.”

Fig. 5 Evolutionary history of the CYP71AZ and CYP83F subfamily in Apiaceae, Rutaceae, Moraceae and Fabaceae. (A) Phylogenetic tree of the CYP71s family. (B) The syntenic relationship of *CYP71AZ18* and *CYP71AZ19* with homologous genes in *A. sinensis*, *P. praeruptorum* and *P. sativa*. Red lines connect homologs and *CYP71AZ18* between any two syntenic regions.

“The FC biosynthesis pathway represents a compelling case of convergent evolution in higher plants, suggesting that distinct enzyme families may have been independently recruited in different plant families to catalyze the same reactional steps^{21,49}. *CYP71AZ18* catalyzes hydroxylation at C-5 positions of psoralen, while *CYP71AZ19* and *CYP83F95* react on hydroxylation at C-8 positions, contributing to the final diversity of FCs in *A. dahurica* (Fig. 5). The identification of *CYP71AZ18* fills the sole missing step in the biosynthesis pathways of imperatorin and isoimperatorin, the two standard compounds of *A. dahurica*⁵⁰. Several reports highlighted that lineage-specific tandem

duplications frequently occurred in the CYP71 clan, elucidating that the genetic redundancy ensuing from duplication events serves as a reservoir for functional innovation^{51,52}. Our study revealed that *A. dahurica*-specific proximal duplication might drive copy number expansion and functional diversification of CYP71AZ members (Fig. 5), which might enrich the FC diversity and enable adaptation to local environments or defending biotic stress in *A. dahurica*⁵³. Previous research on the gradual establishment of the complex coumarin biosynthetic pathway in Apiaceae suggested that late-origin might explain the limited distribution of complex coumarin-containing taxa restricted to derived species in Apiaceae¹⁵. Therefore, additional metabolomic and genetic data are required to completely elucidate origin and evolution of FC biosynthesis pathway in Apiaceae. Moreover, given the convergent evolution of FC biosynthesis and functional convergence observed between CYP71AZ19 and CYP83F95, it is essential to adopt broad thresholds in candidate gene screening based on phylogenetic clades. Lineage-specific evolutionary analyses, transcriptomic assays, biosynthetic gene cluster identifications, and stress response studies, are complementary strategies in candidate gene screening.”

Comment 3: As enriched in special volatile terpenoids, A. dahurica genome should have footprints of the special evolutionary trajectory of related pathways and gene evolution, which is worthy of further digging of the genome data and transcriptome data for more indications or markers.

Response: Thank you for your suggestion! we consider it a promising avenue for further investigation. Our previous study, “*Widely targeted metabolomics analysis reveals differences in volatile metabolites among four Angelica species*”, identified 190 volatile terpenoid metabolites in the roots of *A. dahurica*³. This makes *A. dahurica* an excellent model for studying terpenoid biosynthesis pathways. Notably, in the Moraceae, the marmesin synthase CYPF12 has been linked to terpenoid metabolism. It would be interesting to explore whether enzymes involved in the FC biosynthesis pathway in *A. dahurica* exhibit similar functions in terpenoid biosynthesis. For the sake of keeping the theme focused on coumarin biosynthesis in this manuscript, we chose not to include the analysis, but we highlight it as a potential research direction in the discussion section (line 435-446, page 16). It now reads,

“In conclusion, our study provides a valuable genomic resource of *A. dahurica*, enriches the secondary metabolite profile during its root development, fills an enzyme gap in the FC biosynthetic pathway, explores the evolution of the FC biosynthetic genes, and dissects the epigenetic regulation of gene expression and metabolite content. This research advances our knowledge of the FC biosynthesis pathway and the role of lineage-specific duplication of CYP450 in diversifying FCs, and contributes to the understanding of the impact of epigenetic regulation on gene expression, which provides insights into metabolic production of FCs via biosynthetic technology. Additionally, we provide genomic, transcriptomic, metabolomic, and ATAC-seq data, which serve as valuable resources for future studies on the biosynthesis and evolution of enzymes involved in other important metabolites in *A. dahurica*, such as volatile terpenes.”

- 1 Hunziker, P. *et al.* Herbivore feeding preference corroborates optimal defense theory for specialized metabolites within plants. **118**, e2111977118 (2021).
- 2 Alba, C., Bowers, M. D. & Hufbauer, R. J. E. Combining optimal defense theory and the evolutionary dilemma model to refine predictions regarding plant invasion. **93**, 1912–1921 (2012).
- 3 Ji, J. *et al.* Widely targeted metabolomics analysis reveals differences in volatile metabolites among four *Angelica* species. *Nat Prod Bioprospect* **15**, 2, doi:10.1007/s13659-024-00485-5 (2025).

Response to reviewers

Dear Editor,

Thank you for considering our manuscript, “Integrative multi-omics data provide insights into the biosynthesis of furanocoumarins and mechanisms regulating their accumulation in *Angelica dahurica*”. We gratefully thank the editor and the three reviewers for their time spent on these constructive comments. We have revised the typos pointed out by Reviewer #2 and further amended the manuscript in accordance with the attached document. Below the comments of the reviewers are our point-by-point responses.

Thank you kindly,

Dr. Li Wang

Reviewer #1

Comments to the Author:

The authors have addressed all my concerns. I am pleased to recommend this excellent work for prompt publication in its current form.

Response: Thank you! We are very grateful for the comments.

Reviewer #2

Comments to the Author:

The authors have made extensive corrections, as requested by the 3 reviewers. The new version of the manuscript reads well and is ready for publication provided that a few typos are corrected:

L361: changed insectsthis by insects

L952: missing final n in Shenzhe?

Response: We sincerely appreciate your careful review! We have corrected the typos and carefully reviewed the manuscript.

Reviewer #3

Comments to the Author:

*The revised manuscript has been appropriately improved, and the authors addressed my concerns. Now, I think it is a nice work with an in-depth demonstration of biosynthesis, regulation, and evolutionary mechanisms for furanocoumarins in *Angelica dahurica*, in comparison with other *Angelica* leaneages.*

The work now presents significant advances in understanding of FC biosynthesis and evolution based on multi-omics data, also containing solid biochemical evidence for CYP71AZ18-catalyzed hydroxylation at C-5 position of psoralen and CYP71AZ19- and CYP83F95-catalyzed hydroxylation at C-8 position to produce xanthotoxol.

Response: Thank you! We are very grateful for the comments.